# Intracellular “In Silico Microscopes”—Comprehensive 3D Spatio-Temporal Virus Replication Model Simulations

**DOI:** 10.3390/v16060840

**Published:** 2024-05-24

**Authors:** Markus M. Knodel, Arne Nägel, Eva Herrmann, Gabriel Wittum

**Affiliations:** 1Simulation in Technology, TechSim, 75248 Ölbronn-Dürrn, Germany; 2Modular Supercomputing and Quantum Computing (MSQC), Goethe-Universität Frankfurt, 60325 Frankfurt am Main, Germany; naegel@em.uni-frankfurt.de; 3Institute for Biostatistics und Mathematical Modelling (IBMM), Goethe-Universität Frankfurt, 60590 Frankfurt am Main, Germany; herrmann@med.uni-frankfurt.de; 4Modelling and Simulation (MaS), Computer, Electrical and Mathematical Science and Engineering (CEMSE), King Abdullah University of Science and Technology (KAUST), Thuwal 23955-6900, Saudi Arabia; gabriel.wittum@kaust.edu.sa

**Keywords:** virus modeling, 3D simulations, diffusion–reaction PDEs, realistic reconstructed geometries, unstructured grids, interdisciplinary virus research, coupling manifold and volume effects

## Abstract

Despite their small and simple structure compared with their hosts, virus particles can cause severe harm and even mortality in highly evolved species such as humans. A comprehensive quantitative biophysical understanding of intracellular virus replication mechanisms could aid in preparing for future virus pandemics. By elucidating the relationship between the form and function of intracellular structures from the host cell and viral components, it is possible to identify possible targets for direct antiviral agents and potent vaccines. Biophysical investigations into the spatio-temporal dynamics of intracellular virus replication have thus far been limited. This study introduces a framework to enable simulations of these dynamics using partial differential equation (PDE) models, which are evaluated using advanced numerical mathematical methods on leading supercomputers. In particular, this study presents a model of the replication cycle of a specific RNA virus, the hepatitis C virus. The diffusion–reaction model mimics the interplay of the major components of the viral replication cycle, including non structural viral proteins, viral genomic RNA, and a generic host factor. Technically, surface partial differential equations (sufPDEs) are coupled on the 3D embedded 2D endoplasmic reticulum manifold with partial differential equations (PDEs) in the 3D membranous web and cytosol volume. The membranous web serves as a viral replication factory and is formed on the endoplasmic reticulum after infection and in the presence of nonstructural proteins. The coupled sufPDE/PDE model was evaluated using realistic cell geometries based on experimental data. The simulations incorporate the effects of non structural viral proteins, which are restricted to the endoplasmic reticulum surface, with effects appearing in the volume, such as host factor supply from the cytosol and membranous web dynamics. Because the spatial diffusion properties of genomic viral RNA are not yet fully understood, the model allows for viral RNA movement on the endoplasmic reticulum as well as within the cytosol. Visualizing the simulated intracellular viral replication dynamics provides insights similar to those obtained by microscopy, complementing data from in vitro/in vivo viral replication experiments. The output data demonstrate quantitative consistence with the experimental findings, prompting further advanced experimental studies to validate the model and refine our quantitative biophysical understanding.

## 1. Introduction

### 1.1. Biological Basics

Virus endemics and pandemics pose significant threats to worldwide healthcare management, economic prosperity, equitable distribution of basic goods, democratic order, and the existence of both humans and animals [1,2,3,4,5,6].

To adequately prepare for future pandemics, it is crucial to explore the fundamental principles of virus replication using modern natural science tools.

The well-known SARS-CoV-2 virus, which belongs to the family of plus-stranded RNA viruses [7], exhibits a relatively simple virus replication strategy. This characteristic has facilitated groundbreaking approaches to understanding virus replication.

Similarly, the hepatitis C virus (HCV), which also belongs to the family of plus-stranded RNA viruses, is one of the major causes of liver transplants in western countries [5,8]. Over decades, the HCV has been extensively studied by experimental virologists [9,10,11], leading to the development of direct antiviral agents capable of permanently eliminating the HCV from patients’ blood [12,13,14]. This remarkable achievement is due to extensive experimental research, which has provided a significant qualitative understanding of virus dynamics. However, there is a lack of basic quantitative biophysical understanding and detailed experiments. To support clinical and virological research, kinetic mathematical models for HCV infection have been developed [15,16,17,18,19,20,21,22] using ordinary differential equation (ODE) models. These models enable the simulation of treatment effects and aid in optimizing the dosing of antiviral agents.

While some viral kinetic studies incorporate the spatial effects of virus replication [23,24,25,26,27,28,29], there remains an untapped potential in leveraging available experimental data for more detailed mathematical models and numerical simulations. In particular, spatially resolved experimental data from virological research can be integrated with spatial model simulations at the intracellular level. For instance, in the case of the HCV, the virus genome replication factories are situated within intracellular structures known as the membranous web (MW). Experimental data on the spatial structure of the membranous web and endoplasmic reticulum (ER) are available [11,30]. The membranous web serves as an accumulation zone for nonstructural HCV proteins (NSPs), forming atop the surface of the ER after infection. The ER itself constitutes an intracellular network with a tubular interconnected structure, where the ER surface encloses the ER lumen. Mathematically, the ER surface represents a system of connected tori with a total number *g* where g≥1. Thus, the ER surface manifests as a 3D embedded curved 2D manifold and acts as the exterior boundary of the ER lumen, which is a connected 3D volume set.

Currently, spatially resolved fluorescence and electro-tomography data are not commonly integrated into computational models of the HCV viral genome replication cycle or those of other viruses. This limitation stems from the inability of ordinary differential equation models to account for spatial structures. However, partial differential equation (PDE) models allow for spatially and temporally resolved numerical simulations.

### 1.2. Aims of this Study

The primary aim of this study is to establish a framework capable of simulating quantitatively reliable mathematical models that describe the biophysics of virus replication cycles in a fully spatio-temporal manner, without targeting a specific biological question at this stage. This framework facilitates the interpolation and extrapolation of spatial data from in vitro and in vivo virus replication experiments, enabling the prediction of the spatial patterns of the components driving virus replication. Moreover, it provides a platform for unveiling biological questions “on the fly” as demonstrated in this study for some specific issues.

Specifically, this study aims to extend previous model approaches [31,32,33] by combining the dynamics of the intracellular HCV replication cycle, which is restricted to the ER surface, with other aspects of the viral replication cycle occurring in the cell cytosol and membranous web. Concerning the effects on the ER surface, this includes considering the concentrations of various components, such as ribosomal bound viral RNA, viral polyprotein translated at ribosomes, nonstructural protein cleaved from the polyprotein, and polymerized free viral RNA attached to the ER surface. Furthermore, the effects occurring in the volumes of the cytosol and membranous web involve the concentrations of components of nonstructural proteins detached from ribosomes to form the membranous web, nonstructural proteins incorporated into the membranous web, replication complexes, polymerized free viral RNA moving in the cytosol, and a generic host factor.

The model simulations allow viral RNA (vRNA) to move in the full volume (cytosol and membranous web) and attach to the ER surface to move—attached to this surface—on it further. However, the movement of the generic host factor is confined to the volume (cytosol and membranous web), reflecting the assumption that this factor represents cell energy essential for virus replication and should not be restricted to the ER surface. Finally, while the movement of nonstructural proteins is limited to the ER surface, as indicated by experiments, the accumulation of the membranous web zones occurs in the full volume of the membranous web regions, where these volumes are separated from the rest of the cell.

Previous studies were divided into two categories: qualitative models of the full genome replication cycle [31,32] using heuristic diffusion and reaction parameters of the model components—and experimental data-based estimations of diffusion coefficients for single parameters and components [33,34], suitable for full genome replication cycle models. In this study, we combined both approaches to achieve biophysical model simulations that accurately reflect experimental reality in a quantitatively reliable manner. The kinetic coefficients were drawn from previous modeling efforts or experimental work [19,33]. All simulations were performed using geometries reconstructed from experimental data [30,31,32].

### 1.3. Organization of this Paper

The forthcoming chapters of this article will explore our in silico modeling approach. It is noteworthy that while this study builds on our previous studies [31,32,33,34], it is self-contained. There is no need to review previous studies to understand the content of this study.

This paper is organized as follows:

In Section 2 (“Materials, Models and Methods”), we present a novel mathematical model of the viral replication cycle. We explore the geometric base of our simulations while also providing insights into the solution techniques. Leveraging modern supercomputers, we solve the computational model.

Section 3 (“Results”) presents the simulations based on our model and describes the computational results in a biological context.

In Section 4 (“Discussion”), we explore the implications of our model in the context of modern experiments. Furthermore, we analyze the quantitative reliability of our simulations by comparing them with experimental data.

Finally, Section 5 (“Conclusions”) summarizes the key findings of our study and offers an outlook on the medium- and long-term perspectives of our framework in virus research, including its integration with in silico–in vitro research.

The appendices provide additional details that are not required to understand the major aspects of the study but might be helpful for a broader understanding.

We highlight that the primary results and discussions in Section 3 and Section 4 should remain comprehensible and coherent even without delving into the technical and mathematical details of Section 2. We recommend that readers who are more inclined toward the experimental aspects avoid Section 2 during their initial reading.

## 2. Materials, Models and Methods

In this section, we present a mathematical model of the HCV viral RNA (vRNA) replication cycle and describe the technical basics of its evaluation on a cutout part of a realistically reconstructed hepatoma cell geometry, as shown in Figure 1.

We begin by describing the establishment of the unstructured grid based on experimental fluorescence data.

Following this, we develop a complex nonlinear mathematical model of the viral RNA cycle that integrates the surface and volume effects of the major components of the viral RNA cycle.

Finally, we discuss the numerical solution techniques used for conducting in silico simulations on the present supercomputers. While we provide a high-level overview of the mathematical and numerical properties in this section, for a more detailed description, we refer readers to a peer-reviewed proceedings paper [35] and to another peer-reviewed proceedings paper that is currently under review [36].

### 2.1. Computational Domain: Experimental Data Based Unstructured Grid Geometry

The aim of this section is to introduce the computational domain and its subdomains relevant for the partial differential equation (PDE) model, which we will introduce in Section 2.3 below.

The geometries used as the basis for the simulations are based on the surfaces of the ER and membranous web, reconstructed from experimental data [30].

These surface meshes have been presented in our previous studies [31,32], where we provided detailed insight into the technical aspects of the reconstruction and unstructured grid generation processes. However, in this study, understanding the technical steps of the reconstruction process is not essential; it is sufficient to assume the triangulated surface grids of ER and membranous webs are given. However, for the interested readership, Section A.1 revisits the major issues of the reconstruction process.

The starting point of this study is the given mesh describing a cutout part of a hepatoma cell, describing the surfaces of the ER and membranous webs within this cutout part, as illustrated in Figure 1A.

The central challenge of this study, in contrast to previous studies, is the incorporation of volume effects, which necessitates the establishment of a volume grid [35].

We extend the pure surface grid into a surrounding hexahedron and tetrahedralize the volume. This results in a mathematical representation of the volume and ER surface of a cutout part of the hepatoma cell, which can be used to simulate the viral replication cycle. Specifically, the ER is represented by a triangulated surface, with ribosomes interpreted as intersections between the ER and membranous web surfaces. The membranous web is modeled as a volume region within its surface. The remaining volume is considered to be the cytosol, except for the ER lumen, which is omitted from the volume mesh because it does not participate in the genome replication cycle. Figure 1B,C illustrate the enclosing of the surface grid of the ER and membranous webs by the hexahedron, while Figure 1D–F show the various subdomains of the merged surface and volume grid.

Table 1 defines the technical symbols and their biological interpretations for the various subdomains within the surface and volume grids, i.e., the subdomains of the 3D computational domain Ω=Γ∪C∪W. The external hexahedron is denoted as B, with ∂Ω=Γ representing the external boundary of $\Omega \subset R^{3}$. The ER surface/manifold M is embedded within C∪W, but the ER lumen described by the volume enclosed by M is excluded from the computational domain and not meshed. Therefore, Ω is not stellated, but M is entirely part of the “external” boundary of Ω, i.e., Γ=M∪B.

The generated mesh serves as the computational domain of our simulations, representing grid refinement level 0 in the context of global grid refinement strategies.

Further details (concerning surface grid generation) can be found in Section A.1, with additional information available in our previous publications [31,32,35].

### 2.2. Tangential Space and Manifold Differential Operators

In accordance with the principles of differential geometry [37], we employed tangential differential operators for the mathematical operations on the manifold: the nabla/gradient operator $\nabla_{T}$ and the divergence operator div_T_. It is worth noting that we labeled these operators with the “index” “T” to signify their projection onto the tangential space of the embedded manifold. Following the approach described by [38] for FE discretizations [39] and adapting it for vertex-centered finite volume discretizations [40,41], our computations were performed using Cartesian coordinates. At each element of the discretized manifold (which corresponds to a face of the volume grid, typically a triangle in our case), the differential operators were projected from the full 3D space onto the tangential space of the manifold, i.e., the 2D hypersurface M, more in detail onto the corresponding 2D surface/manifold element (triangle), which is embedded into the complete 3D space.

The standard 3D differential operator is as follows: $\partial_{i}=\frac{\partial }{\partial x_{i}}$. Using Einstein summation convention, we express the differential operator $\partial_{T}$ tangential to the manifold (using indices)
(1)$${\left(\partial_{i}\right)}_{T}=\partial_{i}-{\vec{n}}_{i}{\vec{n}}_{j}\partial_{j}=\partial_{i}-{\vec{n}}_{i}<\vec{n},\vec{\partial }>\;\mathrm{respectively},\;\partial_{T}=\partial -\vec{n}<\vec{n},\partial >$$
without indices, where $\vec{n}$ is that vector which is normal to the respective element (triangle) of the manifold. Defining the projector
(2)$$P=1\;l-\vec{n}⊗\vec{n},\;\;\mathrm{we}\;\mathrm{obtain}\;\;\nabla_{T}u=\nabla u-\nabla u\cdot \vec{n}⊗\vec{n}=\nabla u-\left(\nabla u\cdot \vec{n}\right)\cdot \vec{n}$$
as the tangential part $\nabla_{T}$ of the gradient operator ∇ applied to a concentration *u*. For the tangential part div_T_ of div, an analogous result arises.

### 2.3. Mathematical Partial Differential Equation (PDE) Model Coupling Surface and Volume Effects

The model developed in this study follows the structure introduced in [35], where the focus was on numerical solution techniques rather than biophysically relevant results. However, compared with the published coupled surface-volume model in [35], our model has been slightly improved, particularly in terms of specific coefficients of diffusion and reaction to better replicate the experimental reality. In modeling the concentrations of the major compartments of the viral RNA replication cycle, we considered their spatial location and mode of action. We defined concentrations residing solely on the manifold and others residing solely in the volume.

A comprehensive list of all compartments (concentrations) is provided in Table 2, which describes various forms of viral RNA (vRNA), nonstructural viral proteins (NSP), and a generic host factor. The non structural proteins are assumed to be the cleavage products of the viral polyprotein translated by the viral RNA.

Specifically, we only considered a generic membranous web-forming web protein (WP) and the NS5A nonstructural protein [9,11,12,42,43,44]. The replication complex (RC) was modeled as a combination of the web protein and viral RNA.

A simplified model scheme that summarizes the main components of the PDEs is shown in Figure 2.

The mathematical diffusion–reaction model is based on the coupling of surface partial differential equations (sufPDEs), partial differential equations (PDEs), and connecting boundary flux conditions of the PDEs, which are reflected by the reaction terms of the sufPDEs. The highly nonlinear structure of the diffusion and reaction coefficients follows the principles introduced in [32] and reflects the central properties elucidated therein.

The coefficient structure is inspired by Michaelis–Menten kinetics. Assuming c≥0 to be a generic concentration of the model and *f* a strictly positive constant characterizing the “kinetics” of interest, reaction terms of the form r(c)=c/(c+f) allow for a relatively sharp increase in the values of r(c) as *c* increases from 0, and for “plateaux”, which is the value defining the horizontal asymptote of each reaction term of the type r(c). When f=0, the term r(c)=1 remains constant. We use this feature in certain cases, when reactions are constructed following the principle $\bar{r}\left(c\right)=\prod_{i=0}^{n}\frac{c_{i}}{c_{i}+f_{i}}$ with at least one $f_{i}\neq 0$, i=1,…,n while some of the $f_{i}$ may be zero, although it is not a necessity. This feature enhances the model’s flexibility.

Defining $\chi_{R_{i}}$ as the characteristic function of $R_{i}$, i.e.,
(3)$$\chi_{R_{i}}\left(x\right)=\left\{\begin{array}{cc} 1, & \mathrm{if}x\in R_{i},\\ 0, & \mathrm{if}x\notin R_{i}, \end{array}\right.$$
we use the following abbreviations:(4)$$\begin{array}{ccc} \eta_{0}\left(x\right) & = & \frac{1}{\sum_{i=1}^{7}\left|R_{i}\right|\chi_{i}\left(x\right)+\delta }\left[\sum_{j=1}^{7}\chi_{j}\left(x\right)\right] \end{array}$$
where $|R_{i}|$ represents the size of the surface of the ribosomal region $R_{i}$, i=1,2,…,7, and δ is a constant damping value given in μm^2^ to be chosen. The values used in this study are shown in Table 3.

The notation of all equations does not specify the spatial and temporal reference points for all concentrations as it remains consistent across all concentrations. Henceforth, we denote $\eta_{0}$ without explicitly referring to the spatial point *x*.

To formulate the PDE (partial differential equation) system, we incorporate several parameters that are listed in Table 3 along with their numerical values and units. Parameters with units are $r_{i}\in R^{+}$, i=1,2,…,6, which describes reaction processes and has the unit 1/s (we assume $r_{5}<r_{3}$), the diffusion coefficients $D_{R},D_{P},D_{N},D_{R}^{S},D_{C},D_{R}^{V},D_{H}\in R^{+}$ given in the unit $\frac{\mu m^{2}}{s}$ and the initial value of the host factor concentration $h_{0}$ with unit $\frac{1}{\mu m^{3}}$. The remaining parameters are dimensionless numbers: Coefficients $p_{i}\in R^{+}$, i=1,…,10, the coefficients $k_{i}\in R^{+}$ for i=1,…,3 and $v_{1},v_{2}\in N^{+}$ as well as $b\in R^{+}$. Furthermore, we use ν∈N.

Finally, we define reaction the terms
(5a)$$\begin{array}{c} \begin{array}{c} \begin{array}{cc} R_{C} & =r_{1}R_{R}^{S}\frac{R_{R}^{S}}{R_{R}^{S}+p_{1}}\frac{W_{C}^{S}}{W_{C}^{S}+p_{2}}\frac{H^{V}}{H^{V}+p_{9}} \end{array} \end{array} \end{array}$$
(5b)$$\begin{array}{c} \begin{array}{c} \begin{array}{cc} R_{W} & =r_{4}W_{C}^{S} \end{array} \end{array} \end{array}$$
(5c)$$\begin{array}{c} \begin{array}{c} \begin{array}{cc} R_{N} & =r_{5}N_{E}^{S}W_{C}^{S} \end{array} \end{array} \end{array}$$
(5d)$$\begin{array}{c} \begin{array}{c} \begin{array}{cc} R_{R} & =r_{6}R_{P}^{V} \end{array} \end{array} \end{array}$$
and flux terms
(6a)$$\begin{array}{c} \begin{array}{c} \begin{array}{cc} J_{C} & =D_{C}\frac{W_{W}^{V}}{W_{W}^{V}+p_{6}}\nabla C_{W}^{V} \end{array} \end{array} \end{array}$$
(6b)$$\begin{array}{c} \begin{array}{c} \begin{array}{cc} J_{W} & =D_{N}\nabla W_{W}^{V} \end{array} \end{array} \end{array}$$
(6c)$$\begin{array}{c} \begin{array}{c} \begin{array}{cc} J_{N} & =b\cdot D_{N}\frac{W_{W}^{V}}{W_{W}^{V}+p_{5}}\nabla N_{W}^{V} \end{array} \end{array} \end{array}$$
(6d)$$\begin{array}{c} \begin{array}{c} \begin{array}{cc} J_{R} & =D_{R}^{V}\left(1+k_{2}\frac{N_{W}^{V}}{N_{W}^{V}+p_{7}}\right)\nabla R_{P}^{V} \end{array} \end{array} \end{array}$$With these definitions, we can establish the coupled sufPDE/PDE system as follows:

The sufPDEs (surface partial differential equations) are as follows:
(7a)$$\begin{array}{ccc} \partial_{t}R_{R}^{S} & ={\mathrm{div}}_{T}\left(D_{R}^{S}\nabla_{T}R_{R}^{S}\right)-\underset{R_{C}}{\underset{︸}{r_{1}R_{R}^{S}\frac{R_{R}^{S}}{R_{R}^{S}+p_{1}}\frac{W_{C}^{S}}{W_{C}^{S}+p_{2}}\frac{H^{V}}{H^{V}+p_{9}}}}+r_{8}R_{E}^{S}\left(\nu \eta_{0}-R_{R}^{S}\right)\; & \;\forall \vec{x}\in R \end{array}$$
(7b)$$\begin{array}{ccc} \partial_{t}P_{R}^{S} & ={\mathrm{div}}_{T}\left(D_{P}\nabla_{T}P_{R}^{S}\right)+r_{2}R_{R}^{S}\frac{H^{V}}{H^{V}+p_{3}}-r_{3}P_{R}^{S}\; & \;\forall \vec{x}\in R \end{array}$$
(7c)$$\begin{array}{ccc} \partial_{t}W_{C}^{S} & ={\mathrm{div}}_{T}\left(D_{N}\nabla_{T}W_{C}^{S}\right)+r_{3}P_{R}^{S}-\underset{R_{W}}{\underset{︸}{r_{4}W_{C}^{S}}}-\underset{v_{1}R_{C}}{\underset{︸}{v_{1}r_{1}R_{R}^{S}\frac{R_{R}^{S}}{R_{R}^{S}+p_{1}}\frac{W_{C}^{S}}{W_{C}^{S}+p_{2}}\frac{H^{V}}{H^{V}+p_{9}}}}\; & \;\forall \vec{x}\in R \end{array}$$
(7d)$$\begin{array}{ccc} \partial_{t}N_{E}^{S} & ={\mathrm{div}}_{T}\left(D_{N}\nabla_{T}N_{E}^{S}\right)+{\left[r_{3}P_{R}^{S}-\underset{R_{N}}{\underset{︸}{r_{5}N_{E}^{S}W_{C}^{S}}}\right]}_{R}\; & \;\forall \vec{x}\in M \end{array}$$
(7e)$$\begin{array}{ccc} \partial_{t}R_{E}^{S} & ={\mathrm{div}}_{T}\left[D_{R}^{S}\left(1+k_{1}\frac{N_{E}^{S}}{N_{E}^{S}+p_{4}}\right)\nabla_{T}R_{E}^{S}\right]+\underset{R_{R}}{\underset{︸}{r_{6}R_{P}^{V}}}-{\left[r_{8}R_{E}^{S}\left(\nu \eta_{0}-R_{R}^{S}\right)\right]}_{R}\; & \;\forall \vec{x}\in M \end{array}$$

The “volume” PDEs (PDE/volPDE) ((volume) partial differential equations) are as follows:
(8a)$$\begin{array}{ccc} \partial_{t}W_{W}^{V} & =\mathrm{div}\underset{J_{W}}{\underset{︸}{\left(D_{N}\nabla W_{W}^{V}\right)}}\; & \;\forall \vec{x}\in W \end{array}$$
(8b)$$\begin{array}{ccc} \partial_{t}N_{W}^{V} & =\mathrm{div}\underset{J_{N}}{\underset{︸}{\left(b\cdot D_{N}\frac{W_{W}^{V}}{W_{W}^{V}+p_{5}}\nabla N_{W}^{V}\right)}}\; & \;\forall \vec{x}\in W \end{array}$$
(8c)$$\begin{array}{ccc} \partial_{t}C_{W}^{V} & =\mathrm{div}\underset{J_{C}}{\underset{︸}{\left(D_{C}\frac{W_{W}^{V}}{W_{W}^{V}+p_{6}}\nabla C_{W}^{V}\right)}}\; & \;\forall \vec{x}\in W \end{array}$$
(8d)$$\begin{array}{ccc} \partial_{t}R_{P}^{V} & =\mathrm{div}\left[\underset{J_{R}}{\underset{︸}{D_{R}^{V}\left(1+k_{2}\frac{N_{W}^{V}}{N_{W}^{V}+p_{7}}\right)\nabla R_{P}^{V}}}\right]+{\left[r_{7}C_{W}^{V}\frac{H^{V}}{H^{V}+p_{8}}\frac{W_{W}^{V}}{W_{W}^{V}+p_{10}q}\right]}_{W}\; & \;\forall \vec{x}\in \Omega \end{array}$$
(8e)$$\begin{array}{ccc} \partial_{t}H^{V}\; & =\mathrm{div}\left[D_{H}\left(1+k_{3}\frac{W_{W}^{V}}{W_{W}^{V}+p_{9}}\right)\nabla H^{V}\right]-{\left[v_{2}r_{6}C_{W}^{V}\frac{H^{V}}{H^{V}+p_{8}}\frac{W_{W}^{V}}{W_{W}^{V}+p_{10}}\right]}_{W}\; & \;\forall \vec{x}\in \Omega \end{array}$$

The boundary conditions of the volPDEs establish a connection between the reaction terms of the sufPDEs and the external boundary fluxes of the volume components. This connection ensures strict mass conservation by accounting for the interchange of components between the manifold and volume. Equations for boundary conditions establish the relationship between flux conditions at the boundaries of the volume and the reactions on the surface, thereby ensuring the conservation of mass in the exchange between the manifold and volume:
(9a)$$\begin{array}{cccc} \vec{n}\cdot \left[\underset{J_{W}}{\underset{︸}{D_{N}\nabla W_{W}^{V}}}\right]\; & =+\underset{R_{W}}{\underset{︸}{r_{4}W_{C}^{S}}}\; & \; & \forall \vec{x}\in R \end{array}$$
(9b)$$\begin{array}{cccc} \vec{n}\cdot \left[\underset{J_{N}}{\underset{︸}{b\cdot D_{N}\frac{W_{W}^{V}}{W_{W}^{V}+p_{5}}\nabla N_{W}^{V}}}\right]\; & =+\underset{R_{N}}{\underset{︸}{r_{5}N_{E}^{S}W_{C}^{S}}}\; & \;\; & \forall \vec{x}\in R \end{array}$$
(9c)$$\begin{array}{cccc} \vec{n}\cdot \left[\underset{J_{C}}{\underset{︸}{D_{C}\frac{W_{W}^{V}}{W_{W}^{V}+p_{6}}\nabla C_{W}^{V}}}\right]\; & =+\underset{R_{C}}{\underset{︸}{r_{1}R_{R}^{S}\frac{R_{R}^{S}}{R_{R}^{S}+p_{1}}\frac{W_{C}^{S}}{W_{C}^{S}+p_{2}}\frac{H^{V}}{H^{V}+p_{9}}}}\; & \;\; & \forall \vec{x}\in R \end{array}$$
(9d)$$\begin{array}{cccc} \vec{n}\cdot \left[\underset{J_{R}}{\underset{︸}{D_{R}^{V}\left(1+k_{2}\frac{N_{W}^{V}}{N_{W}^{V}+p_{7}}\right)\nabla R_{P}^{V}}}\right]\; & =-\underset{R_{R}}{\underset{︸}{r_{6}R_{P}^{V}}}\; & \;\; & \forall \vec{x}\in M \end{array}$$

In essence, the boundary flux conditions of the PDEs are mirrored by the reactions of the sufPDE system.

The reaction terms of the sufPDE system R and their corresponding boundary condition terms of the partial differential equation (PDE) system, denoted as J, share the same indices (C,W,N,R).

Section A.2 presents the complete sufPDE/PDE system using alternative representation forms to accommodate different reader preferences.

The initial conditions are as follows:(10a)$$\begin{array}{c} \begin{array}{c} \begin{array}{cc} R_{R}^{S}\left(\vec{x},t=0\right) & =\left\{\begin{array}{cc} \eta_{0}\left(\vec{x}\right)\; & \forall \;\vec{x}\in R_{2}\\ 0 & \forall \;\vec{x}\in M\setminus R_{2} \end{array}\right. \end{array} \end{array} \end{array}$$
(10b)$$\begin{array}{c} \begin{array}{c} \begin{array}{cc} P_{R}^{S}\left(\vec{x},t=0\right) & =0\;\forall \;\vec{x}\in M \end{array} \end{array} \end{array}$$
(10c)$$\begin{array}{c} \begin{array}{c} \begin{array}{cc} W_{C}^{S}\left(\vec{x},t=0\right) & =0\;\forall \;\vec{x}\in M \end{array} \end{array} \end{array}$$
(10d)$$\begin{array}{c} \begin{array}{c} \begin{array}{cc} N_{E}^{S}\left(\vec{x},t=0\right) & =0\;\forall \;\vec{x}\in M \end{array} \end{array} \end{array}$$
(10e)$$\begin{array}{c} \begin{array}{c} \begin{array}{cc} R_{E}^{S}\left(\vec{x},t=0\right) & =0\;\forall \;\vec{x}\in M \end{array} \end{array} \end{array}$$
(10f)$$\begin{array}{c} \begin{array}{c} \begin{array}{cc} W_{W}^{V}\left(\vec{x},t=0\right) & =0\;\forall \;\vec{x}\in \Omega \end{array} \end{array} \end{array}$$
(10g)$$\begin{array}{c} \begin{array}{c} \begin{array}{cc} N_{W}^{V}\left(\vec{x},t=0\right) & =0\;\forall \;\vec{x}\in \Omega \end{array} \end{array} \end{array}$$
(10h)$$\begin{array}{c} \begin{array}{c} \begin{array}{cc} C_{W}^{V}\left(\vec{x},t=0\right) & =0\;\forall \;\vec{x}\in \Omega \end{array} \end{array} \end{array}$$
(10i)$$\begin{array}{c} \begin{array}{c} \begin{array}{cc} R_{P}^{V}\left(\vec{x},t=0\right) & =0\;\forall \;\vec{x}\in \Omega \end{array} \end{array} \end{array}$$
(10j)$$\begin{array}{c} \begin{array}{c} \begin{array}{cc} H^{V}\left(\vec{x},t=0\right) & =h_{0}\;\;\forall \;\vec{x}\in \Omega \end{array} \end{array} \end{array}$$


These conditions ensure that at one selected ribosomal region (we choose $R_{2}$), the integral of the concentration of ribosomal-bound viral RNA in this region starts at one.

Biophysically, this means that at the beginning, there is one bound viral RNA molecule at one ribosomal region, whereas all other concentrations, except for the host factor, start from zero. The host factor itself starts with a constant concentration $h_{0}$, uniformly distributed throughout the cell. In simpler terms, initially, there is one viral RNA in the cell, and the host factor is homogeneously distributed throughout the cell. All other components involved in the viral RNA cycle are absent at time zero but are produced through viral polyprotein translation, cleavage, nonstructural protein movement, and membranous web accumulation, followed by viral RNA polymerization and movement, inducing the closure and repetition of the cycle at other ribosomal regions.

This model encompasses all major steps of the viral RNA replication cycle, with further details to be illustrated in the results section (Section 3).

Section A.3 provides additional examples of coupled-surface volume models from the existing literature.

### 2.4. Michaelis–Menten Kinetics Inspired Reaction Terms

The form of the reaction terms presented in these equations was inspired by Michaelis–Menten kinetics, a structure extensively detailed and validated in [32]. To ensure a comprehensive understanding of this study, we briefly revisit their modeling approach. The structure of a reaction term of coefficient with the form r(c)=c/(c+f), where *c* represents concentration and *f* is an elastic parameter, possesses two major properties. The reaction term has a supremum, which is approached when the concentration *c* increases, whereas it effectively diminishes for small concentrations. Moreover, this structure enables relatively sharp transitions between “off” and “on” contributions of such terms; it remains relatively constant (either zero or one) for a broad range of concentrations. Notably, it prevents any single concentration from dominating all others, even when other concentrations are nearly negligible. However, such dominance may occur in multilinear models of the form r(a,b,c)=fabc for heuristic concentrations a,b.c and *f* being an elastic parameter. Even in the case of nearly vanishing but nonzero values of a,b, a significantly large concentration of *c* enables non-negligible reaction rates, which may not be biophysically plausible. Therefore, multilinear models are often useful only for small concentrations. However, the form inspired by Michaelis–Menten allows for another a broader spectrum of applications.

### 2.5. Numerical Values of the Model Parameters

For the diffusion and reaction coefficients, we take coefficients inspired from the previous modeling [19,33] and experimental work [19]. For the simulation in this study, we used the parameter values shown in Table 3. In addition, we provide some references; however, these may not correspond to exact values but serve as a basis for understanding the size of magnitude of these parameters.

However, our model is not limited to this specific parameter set. Further refinement and calibration of the parameter values are necessary for a more realistic description. In addition, studies are needed to fine-tune the parameters and potentially also refine the model structure itself.

### 2.6. Numerical Solution Techniques for Solving the sufPDE/PDE System

An in-depth description of the numerical solution techniques can be found in [35]. In this context, we provide a brief overview of the key aspects.

Time discretization is achieved using an adaptive implicit Euler scheme of the first order.

For discretization in space, we employed a vertex-centered finite volume (vcFV) scheme, which is also called the “box method” [41,45]. This scheme ensures the conservation of mass for the transported components on at the discrete level.

The sufPDE/PDE system ([Disp-formula FD7a-viruses-16-00840])–(8e) is solved using a nonlinear Newton solver. Each iteration of the Newton solver leads to the formulation of a large system of linear equations (SLE) with a sparse matrix structure. These SLEs are solved using a geometric multigrid solver (GMG) preconditioned with a BiCGStab solver. The GMG uses a hierarchical grid distribution technique that enables highly efficient parallel solutions of the system [46]. For further details on the GMG solver and the numerical grid convergence challenges and solutions for the highly nonlinear model at long-term scales, please refer to [36]. As a smoother, we used symmetric Gauss-Seidel (SGS), and the base solver was a BiCGStab preconditioned with an ILU.

Table 4 provides information on the degrees of freedom (DoFs) at different grid levels.

The computations were performed on an HLRS Stuttgart Apollo Hawk Supercomputer. The final quantitative data evaluations presented in the results section below were obtained through computations at grid refinement level 4. However, the simulation movies Appendix A and the corresponding screenshots are shown at grid refinement level 1 to conserve storage space.

### 2.7. Integrals of Concentrations over Subdomains

During the simulation of the model, the integrals of the concentrations within specific subdomains were evaluated.

For a generic surface concentration $Q^{S}$ defined over a two-dimensional manifold M embedded in the three-dimensional space $R^{3}$, we define the integral over a subset of the manifold S⊂M as follows:(11)$$\begin{array}{c} \begin{array}{c} I_{S}\left(Q^{S}\right):=\int_{S}Q^{S}\;d\sigma . \end{array} \end{array}$$

For a generic 3D volume concentration $Q^{V}$ within the volume over a generic subdomain V⊂Ω, we compute
(12)$$\begin{array}{c} \begin{array}{c} I_{V}\left(Q^{V}\right):=\int_{V}Q^{V}\;d^{3}x. \end{array} \end{array}$$

Given that surface concentrations are provided in units of $\mu m^{2}$ and volume concentrations in $\mu m^{3}$, the integrated values are all dimensionless numbers of components.

Therefore, exchange between volume and manifold is possible without restrictions because of the boundary conditions ([Disp-formula FD9a-viruses-16-00840])–(9d), which are mirrored by the reaction coefficients of the sufPDEs ([Disp-formula FD7a-viruses-16-00840])–(7e).

### 2.8. Distinguishing between Geometrically Defined and Biophysically Active MW Zones

It is important to distinguish between geometric regions, which are subdomains of the computational mesh, as shown in Table 1, and the static and dynamic biophysical effects described by the model. This distinction is particularly crucial for the membranous web. Defined within the computational domain, geometric membranous web zones represent the potentially active regions of the membranous web. However, these zones are pre-existing volume subdomains from the initial setup. They are defined by the dsRNA-stained zones of the experimental z-stack data [30], which form the geometric basis of our simulations. However, at the beginning of the simulations, biophysically, these zones have the same properties as the rest of the cytosol, as the cell is considered to start in a healthy state. A geometric membranous web zone exhibits biophysical activity only when a nonzero spatial concentration of viral web protein (WP) is present within it. This event occurs when ribosomal-bound web protein detaches from the ribosomal zone of the ER, situated adjacent to the geometric membranous web zone. In our model, only web protein dissociating from the ER surface can diffuse into the geometric membranous web zone defined by the experimental z-stack data. Subsequently, the web protein permeates the membranous web zone, making it biophysically “active” as it diffuses into the geometric membranous web volume. Moreover, the detachment of NS5A and RC from the surface is dependent on a pre-activated membranous web, i.e., the presence of a nonzero protein concentration within such a membranous web.

## 3. Results

We simulated the mathematical model introduced in Section 2 (“Materials, Models and Methods”). In this section, we present the spatial data obtained from the simulations. In addition, we provide quantitative evaluations of the integrals of the concentrations of the model compartments. These assessments facilitate comparisons with published in vitro data and ordinary differential equation (ODE) kinetic model values averaged over the entire cell. We currently lack access to spatially resolved quantitative data.

### 3.1. Geometric Basis and Geometric versus Biophysical Subdomains

Simulations were performed using realistic reconstructed grid geometries derived from experimental fluorescence z-stack data [30]. These data enable reconstruction of the ER surface and potential membranous web volume regions. Cytosol volume surrounds the ER and membranous web regions. The viral RNA cycle model does not consider the ER lumen.

We define the cut set of the ER surface and the boundaries (surfaces) of the membranous webs as ribosomal regions. A potential geometric membranous web subdomain initially shares the same biophysical properties as the geometric cytosol subdomain, reflecting the healthy state of the cell.

In our model, non structural proteins (NSP)/web proteins (WP) are not permitted to exist in or migrate to the entire cytosol subdomain. Therefore, the cytosol subdomain always retains its biophysical characteristics. After cleavage from the viral polyprotein, the web protein initially adheres to the ER surface. However, these web proteins can detach from the ER surface and exclusively diffuse into the geometric membranous web subdomain. Consequently, the web protein concentration becomes nonzero within the geometric membranous web subdomain, and this geometric subdomain becomes a biophysically active membranous web zone with the respective biophysical properties.

For more details, please refer to Section 2.8.

### 3.2. Spatial Simulation Data Evaluation—Simulation Movie

We visualize the temporal evaluation of the 3D spatial simulation data by employing a front-view approach at selected time points. This visualization focuses on the different concentrations of the model compartments, as detailed in Table 2. Our 3D simulation results represent the concentrations per unit volume element and per unit surface element. Further conceptual details and extended explanations can be found in Section 4.2.1. “Interpretation of local concentrations” of [32]. However, understanding this study does not depend on prior familiarity with the details. We emphasize that our simulations prioritize spatially resolved concentrations over single particle models such as agent-based random walk models.

#### 3.2.1. Concentrations and Their Spatial Regions

In the fully spatio-temporal simulations, three spatial/geometric compartments play a major role: the ER surface, membranous web volumes, and the cytosol volume.

In the following screenshots, we display all concentrations outlined in Table 2: ribosomal bound viral RNA, viral polyprotein, “web protein” (WP) as cleaved generic nonstructural protein excluding NS5A, NS5A nonstructural protein, free viral RNA, replication complex (RC), and a generic host factor. The nature of this host factor, possibly an energy-giving component of the cell, is not specified in this study.

Certain concentrations are exclusively defined either on the ER surface or within the cytosol volume and/or solely on the geometric potential membranous web subdomains, whereas others are present in both surface and volume modes. In the latter scenario, the exchange between the ER surface and volume occurs under specific conditions as stipulated by the model.

Ribosomal bound viral RNA and polyproteins are confined to the ribosomal zones of the ER surface. Free viral RNA exists in two states: either attached/bound to the ER surface or freely moving over the entire volume. Web proteins are restricted to the ribosomal zone of the ER surface in one mode and can additionally exist in the membranous web volumes in another mode. Similarly, NS5A moves across the entire ER surface in one mode, but it can also exist in the membranous web zones in another mode. The replication complex exclusively exists in the membranous web volume zones, whereas the host factor exists throughout the cytosol and membranous web subdomains.

#### 3.2.2. In Silico Microscopye of the vRNA Cycle

We perceive the simulation data as if observed through an “in silico” microscope. To ensure ease of comparison among screenshots captured at various time points, we maintain a consistent scale for each component throughout the duration at all time points.

The total simulated biophysical time is approximately 7.5 hours (h).

##### Ordering of All Simulation Screenshots

The subfigures show the surface bound and volume concentrations of a single species in one or both modes as indicated. The visualized panels of the different concentrations of the model compartments are identically ordered across all time points of the spatial simulation screenshots (the letters coincide with the subindices of all simulation screenshot figures):ARibosomal bound vRNA (“RNA ribo”) on the ribosomal zones of the ER surface and the replication complex (RC) in the membranous webs (MWs) volume.BPolyprotein on the ribosomal zones of the ER surface.CWeb protein (WP) in the ribosomal zones of the ER surface and membranous web (MW) volumes.DFree viral RNA (vRNA) in the volume and at the ER surface.ENS5A at the ER surface and in the membranous web volumes.FHost factor overall in the volume.

For merged perspectives, ER-located concentrations are shown in all cases, whereas volume concentrations are displayed using an opacity mapping-based rendering procedure. This rendering method ensures that a concentration is displayed when it is above a specific threshold; otherwise, the volume remains transparent. The host factor is displayed using a clip plane that reveals the volume grid with no opacity mapping-based rendering applied, and the volume grid elements are visible. The simulated time is displayed in seconds.

For one of the simulation movies provided in the Appendix A, we exclusively visualized ribosomal vRNA/RC, WP, free vRNA, and host factor. Polyprotein and NS5A are excluded to emphasize the major effects. The other movie encompasses all components, mirroring the content of the screenshots.

##### Initial State: One vRNA Attached to One Ribosomal Zone

At the onset of the simulations, the cell was deemed “healthy”, with the exception of one viral RNA already attached to a ribosomal region, as shown in Figure 3.

##### Viral Protein Production, Movement, and MW Zone Activation

In the initial stages of the simulation, the viral polyprotein is translated from ribosomal-bound viral RNA and remains restricted to the ribosomal zone within the ER surface.

Figure 4 shows a snapshot of the in silico microscope at t≈2 min: The polyprotein cleaves into nonstructural proteins, specifically the web protein and NS5A, in this study. However, our model does not currently distinguish between other specific nonstructural proteins. Movement of the web protein and NS5A remains restricted to the ER surface. Nevertheless, the web protein begins to accumulate, initiating remodeling of the ER surface by detaching from the ribosomal zones and diffusing into the geometrically defined membranous web volume zone. Examining Figure 4 at t≈2 min, we observe that the web protein emerges in the first membranous web zone on top of the first ribosomal region, thus activating this membranous web zone. The concentration of web protein in the volume of the membranous web subdomain indicates the growth of membranous web zones over the ER surface. Therefore, the volume concentration of web proteins in the geometric membranous web zone biophysically “activates” this zone. This accumulation of web protein in volume constitutes a biophysical membranous web zone, fostering growth over the ribosomal region where the translating viral RNA is attached.

The nonlinear structure of our mathematical model ensures that NS5A can detach from the ER surface only once when the membranous web is already biophysically active, i.e., when the concentration of web protein is nonzero inside the membranous web volume. Therefore, NS5A is only visible at the ER surface at this time point. Given that the membranous web zone is relatively new, NS5A is not yet visible within it. The same applies to the construction of the replication complex, making it nearly invisible at this stage. With a very sharp view, it can be weakly detected in the volume. There are even indications that the process of constructing the replication complex and its appearance within this region has begun, albeit at a very low level. Due to the implementation of a concentration based model (rather than e.g., an agent based one), we can detect even minor concentrations at this early stage. However, the biophysical effects of these low concentrations are substantially limited by the Michaelis–Menten model-inspired population dynamics structure of the reaction coefficients. The concentration is still low and does not polymerize viral RNA. Viral RNA is not yet visible in within the volume.

##### vRNA Polymerization and Propagation

Figure 5 shows a screenshot after about t≈10 min.

As the membranous web zone becomes denser, NS5A significantly detaches from the ER surface and enters the membranous web volume. Observation reveals that NS5A diffuses away from the initial ribosomal zone on the ER surface and partly detaches from ribosomes to enter the membranous web zone. The replication complex comprises ribosomal-bound viral RNA and a specific quantity of ribosomal-bound web protein. Both ribosomal bound viral RNA and web protein detach from the ribosomal zones on the ER surface and combine in a defined ratio to form a replication complex within the active membranous web zone volume. Given that the replication complex can exist only in the membranous webs, these webs serve as the centers for viral RNA replication. At this stage, the replication complex becomes visible, visually “covering” the remaining ribosomal bound viral RNA.

Once the first active membranous web accumulates a substantial amount of web protein and NS5A, the replication complex initiates the polymerization of new (free) viral RNA, which can move throughout the entire volume, but with increased “velocity” within the membranous web, attributed to contributions to the diffusion coefficient inside these structures. (We remark that the use of the word “velocity” might be misleading in the context of a diffusion–reaction model of concentrations. We use this word here for illustrative purposes, as the basis of diffusion models is Brownian movement of a huge number of single particles. An enhanced diffusion coefficient corresponds with an enhanced velocity of the single particles undergoing random walk. However, the mathematical structure of concentration based models describing diffusion does not incorporate the single particles under random walk any more. Furthermore, concentration-based models of diffusion are the result of a homogenization procedure of random walk models for a huge number of particles, which was introduced first by Einstein [47]).

We observed the initiation of viral RNA polymerization within the first membranous web zone, with some free viral RNA diffusing into the cell volume within the cytosol, whereas others attach to the ER surface. At this time point, the diffusion of viral RNA away from the membranous web zone where it was produced is already slightly visible. Although free viral RNA can attach to the ER surface, this process is not yet visible so far on the ER surface.

During viral RNA polymerization, the host factor is consumed, resulting in a decrease in the active membranous web zone. However, this effect remains moderate as additional host factors are supplied, partially compensating for the depleted levels and substitutes in part the lacking host factor in the active membranous web zone.

##### Closing of the vRNA Cycle

Some of the newly polymerized viral RNA diffuses from the membranous web into the cytosol, whereas another portion attaches to the ER surface and diffuses away on the manifold (i.e., the ER surface). Therefore, substantial amounts of viral RNA become visible in the cytosol and at the ER surface.

Figure 6 illustrates a screenshot at t≈20 min. It shows attached free viral RNA at the ER surface as well as additional free viral RNA that diffused away and bound to the second ribosomal zone. Once attached to the ER surface, viral RNA can either be recaptured by local ribosomes or diffuse to other ribosomal zones.

Our model is designed such that the total amount of viral RNA per ribosomal zone has a clear and definitive limit. Only a defined natural number, in this case, one, is permitted as the maximum per ribosomal zone in the simulations presented in this study. If an excess of viral RNA accumulates on the ER surface of the ribosomes, it cannot be captured any more and will instead diffuse away.

In our model, the diffusion “speed” of viral RNA on the ER surface is influenced by the NS5A concentration: Viral RNA might be “shuttled” by NS5A. Specifically, the diffusion coefficient is mathematically expressed to be enhanced in the presence of NS5A. However, model parameters can be selected to disable the influence of NS5A on viral RNA diffusion speed. Moreover, the diffusion coefficient is nonzero even when NS5A is not locally present.

Driven by diffusion on the ER surface, some free viral RNA reaches the ribosomal zone adjacent to the initial ribosomal zone, where the first viral RNA was attached, thus neighboring the first active membranous web. This ribosomal zone captures some of the viral RNA previously diffusing freely on the ER surface. The viral RNA binds to the second ribosomal zone, resulting in the translation of new polyproteins. These polyproteins undergo the same process as on the first ribosomal zone: cleaving into web protein and NS5A. Thus, the viral RNA cycle is already closed.

At this stage, the second active membranous web zone is not yet detected, indicating that the concentration of web protein in the volume remains invisible. However, this will change and be discussed in subsequent sections soon.

##### Illumination of MW Spots Like a Domino Effect

Once viral protein production starts in the second ribosomal region, the cleaved web protein detaches from the ribosomes and establishes the next active membranous web zone. Afterwards, also within the second membranous web zone, the replication complex forms, initiating the polymerization of viral RNA.

Figure 7 shows the next activated membranous web zone at t≈40 min.

The host factor is already reduced in the zone of the first active membranous web.

Newly synthesized viral RNA emerging from the second membranous web zone has already reached the third ribosomal zone. The viral RNA is already attached to these ribosomes, inducing membranous web activation, which begins to manifest visually.

With the activation of the next membranous web zone and the translation of additional nonstructural proteins at those ribosomal zones where viral RNA is bound to, the previously observed effects are roughly repeated: accumulation of web protein, entry of NS5A, construction and infiltration of the replication complex, and finally, viral RNA polymerization, which partially diffuses away and partially attaches to the ER surface. On the ER surface, the diffusing free viral RNA reaches the next ribosomal zone, attaches there as ribosomal bound viral RNA. The ribosomal bound RNA translates polyprotein, and the cycle continues…

Figure 8, Figure 9, Figure 10, Figure 11, Figure 12 and Figure 13 display this domino effect-like process unfolding from t≈50 min with intermediate steps until t=7.5 h. In summary, these figures illustrate a series of replication steps. For single events, please refer to the figure legends.

The domino-like cascade causes the “illumination”, i.e., activation of additional membranous web zones followed by viral RNA polymerization within these zones. This polymerization cannot be stopped as long as the host factor is nonzero.

As the replication process progresses, the host factor becomes increasingly depleted, particularly within the already active membranous web zones. The supply of the host factor from the surrounding cytosol leads to a decrease in the host factor not only in the membranous web zones but also within the cytosol itself. Initially, observed locally, this reduction eventually becomes global.

The activation of new membranous web zones continues, which is accompanied by related effects. However, former replication centers experienced a significant reduction in viral RNA polymerization because of host factor depletion. Consequently, viral RNA polymerization nearly ceases within the respective membranous web zones and the viral RNA spots begin to disappear (due to diffusion caused blurring), gradually dissipating in the order of their appearance. Throughout these processes, the membranous web zones, the nonstructural proteins, and the replication complex filling them persist. In addition, the remodeled ER structure is maintained.

After approximately 3 hours, all membranous web zones were activated and contained polymerized viral RNA. Most membranous webs are actively engaged in viral RNA polymerization. However, in some membranous webs, the intense viral RNA spots in the volume have already significantly blurred due to a lack of host factor.

Finally, all viral RNA concentration spots become blurred as the viral RNA disperses in the volume. It diffuses into the cytosol, binds to the ER surface, attaches to ribosomes, and combines with web proteins to form replication complexes.

By the end of the simulated biophysical time, most of the host factor was depleted, whereas most of the viral RNA was located at the ER surface and within the replication complexes inside the membranous webs.

##### Remarks on Visualization Properties

Our visualization approach maintained a fixed scale for each concentration throughout the simulation duration.

We used fixed scales across all graphs to ensure that the displayed concentration $c^{disp}$ remains fixed between 0≤
${\mathrm{const}}_{inf}=c_{inf}<c^{disp}\left(\vec{x},t\right)<c_{sup}=\;$${\mathrm{const}}_{sup}$ for all concentrations of type *c*. Although this approach has the advantages of facilitating comparisons across various states throughout the simulation, it also has drawbacks.

The trade-off lies in the possibility that either (i) $0\leq c\left(\vec{x},t\right)<c^{disp}\left(\vec{x},t\right)$ or (ii) $c\left(\vec{x},t\right)>c^{disp}\left(\vec{x},t\right)$ could occur due to the fixed scaling approach. The restriction of the scale is not an error, and no incorrect results are displayed. However, a fixed scale prevents the graphical visualization of the case that the concentrations are below or above the rendered value scale, i.e., when (i) or (ii) occurs.

Graphical representation does not enable the viewer to decide if the concentration is below the infimum of the displayed scale when $0\leq c_{inf}=c^{disp}\left(\vec{x},t\right)$ occurs, i.e., we might encounter $c_{inf}>c\left(\vec{x},t\right)\geq 0$ when a point is illuminated with the color of the lower scale boundary.

In addition, the graphical representation does not allow us to determine if the concentration surpasses the supremum of the displayed scale when $c^{disp}\left(\vec{x},t\right)=c_{sup}$, i.e., we may have $c\left(\vec{x},t\right)>c_{sup}$ when a point is illuminated with the color of the upper scale boundary.

Nevertheless, these limitations do not diminish the significance of our study. The objective of the simulations is to reproduce the spread of viral components through the cell. The simulation data enable the evaluation of concentrations at each point in space and time without restrictions. Our only limitation lies in the displayed results.

Attempting to comprehensively convey all aspects of our simulations through a few screenshots is an insurmountable challenge. The aim of the graphical representation is to provide insights.

Ultimately, the primary goal of the simulation data is their comparison with the experimental data.

The constrained nature of the graphical representation does not impede the ability to compare simulation data with spatial experimental data, provided that such data are available.

One of the two Appendix A accompanying this study shows the progression of the “in silico microscope” data (i.e., the simulation data) in a compact manner, showcasing the major components for a clearer overview. In contrast, the other movie shows all the components, similar to what is presented in this study.

### 3.3. Quantitative Data Evaluations and Numerical Robustness

To compute the total number of compartments in the model, we integrated the respective concentrations as described in Section 2.7 and the equations provided therein for the manifold (i.e., ER surface) components (Equation 11) and volume components (Equation 12).

The integrated concentrations obtained from the long-term simulations, spanning approximately 7.5 biophysical hours, facilitated comparison with experimentally derived values. This includes the overall amount of viral particles and components, namely nonstructural proteins and viral RNA numbers, as shown in Figure 14.

While the entire biohysical results of this project are exclusively presented in this study, in [36], we investigated the numerical grid convergence behavior of the sum of all nonstructural proteins for the complete long-term simulations conducted in this study. Numerical grid convergence tests belong to standard tests of scientific computing and are a helpful tool to validate the robustness of the computed data. In fact, numerical grid convergence is a necessary condition for the reliability of the computed values. Therefore, this property serves as an important consistency check for highly nonlinear computations. (Even though from a strict theoretical view, it is not a sufficient condition; for highly nonlinear models, it is often the major tool in practical applications, as an extended proof based on numerical analysis techniques might be extremely challenging). A key aspect of the application of this technique was put upon the following relation:

The sum of all forms of web proteins and their combinations must necessarily equal the sum of all forms of NS5A. In brief, the proceedings [36] focusing on numerical methods demonstrate that the sum of all web protein-containing components equals the sum of all NS5A-containing components, as the relative difference between the sums of different forms of NS5A and different forms of web protein decreases with increasing grid refinement level. However, it also demonstrates that for high consistency, high grid refinement levels should be used. Therefore, the values presented in this paper were computed at grid level 4, ensuring high confidence in the generated numerical values [36].

Our computational domain spanned approximately 10–40 μm^3^ volume. At the end of approximately 7.5 hours (of simulated biophysics), the total amount of viral RNA reached approximately 40 (particles), whereas there were approximately 350 nonstructural proteins (particles) in this domain.

Assuming that a typical hepatoma cell covers a volume of about $10^{2}$–$10^{3}\mu$m^3^ [31,32,33,48], the volume of our computational domain represents approximately 1–10% of such a typical cell.

Scaling up the computational domain numbers by the volume upscale factor, we estimate approximately 400–4000 viral RNA and 3500–35,000 nonstructural proteins.

When we upscaled the number of produced viral components to the size of a complete cell, the values aligned with the experimental data and kinetic compartment/ordinary differential equation (ODE) model data [19].

For a detailed discussion and comparison, please refer to the discussion Section 4.6.

### 3.4. Mass Conservation along the Exchange between Manifold and Volume

We performed extensive investigations to ensure that the mass of components is conserved when they interchange between the manifold (ER surface) and the volume.

A detailed demonstration of this conservation is provided in [35].

## 4. Discussion

We presented fully 3D spatially resolved simulations of the HCV RNA cycle. The primary aim of our model was not to address a specific biological question or to fit specific experimental data but to provide a framework for future investigations. Despite this initial limited intention, the biophysical results presented above are quantitatively consistent with data published by other researchers, as discussed in Section 3.3. Furthermore, the results of this study offer additional insights into specific issues, such as efficient transport mechanisms of viral RNA.

In essence, the overarching aim of our framework is to reveal basic properties of the viral replication cycle. This form of understanding enables us to address various problems “dynamically”, which in hindsight, can be regarded as answers to pertinent questions.

Below, we discuss several issues that may be interpreted as answers to biological questions, including insights into potential transport property characteristics of viral RNA.

Simulations were performed within a cutout portion of a realistically reconstructed hepatoma cell based on fluorescence z-stack data of hepatoma cells [30]. Our model integrates processes occurring both on the 3D embedded 2D ER surface manifold and within the full volume of the cytosol and membranous web regions. This model builds on our previous spatial modeling efforts [31,32], which were restricted exclusively to surface effects and incorporated only qualitative effects. In addition, we independently estimated some spatial parameters of such models [33,34] based on experimental FRAP time series for spatial NS5A dynamics [49].

The model simulations presented in this paper integrate the previous approaches and expand on them by fully incorporating volume effects. We extensively investigated the biophysical and numerical properties [35] of the model and solvers to ensure the reliability of the simulated values.

Our model simulates the major steps of the interaction between the key compartments of the HCV RNA replication cycle in a fully spatio-temporal manner. The quantitative evaluation of the results is consistent with published experimental data. However, due to the unavailability of spatially resolved data thus far, we have not been able to validate the spatial patterns of the model with experimental data.

In the following sections, we discuss several properties of the results that we observed, which may be used for model validation and further model improvement.

### 4.1. Model Components and Their Respective Spatial Dynamics

The model incorporates the concentrations of the major compartments of the viral RNA replication cycle, namely viral RNA, two forms of nonstructural proteins, and a generic host factor.

Modeling of spatial properties encompasses two aspects: location patterns and movement dynamics. While some of the components exist solely at the ER surface and others exclusively in the volume, certain components can exist in two modes, capable of movement on the ER surface as well as in the membranous web/or cytosol volume.

The nonstructural proteins were modeled as residing exclusively on the ER surface, whereas the generic host factor was modeled to occupy the entire volume without any explicit surface-restricted mode.

However, the new model enables nonstructural proteins to accumulate and form membranous web zones in the volume, as detailed in Section 4.2.

For viral RNA, both modes at the ER surface and modes in the volume are possible, as discussed in Section 4.3.

All concentrations of the model compartments are assumed to undergo passive transport only. Therefore, the model is based on a diffusion–reaction framework. However, the diffusion and reaction coefficients are carefully selected to enable realistic process modeling, as detailed in Section 4.5.

### 4.2. ER Surface Remodeling/MW Zone Establishment

Our model considers the fact that each type of nonstructural protein (WP and NS5A) attaches to the ER surface immediately after their cleavage, with their movement restricted to the ER surface. However, these nonstructural proteins induce remodeling of the ER surface by clustering together to form membranous web zones, leading to the growth of these membranous web zones.

The growth of biophysically active membranous web zones is described in our model simulations as follows: Initially, the entire volume surrounding the ER surface is considered to act as a functional cytosol.

Once cleaved, web proteins can detach from the ER surface and move into the potential membranous web volume subdomain. The potential membranous web subdomain is part of the volume and is defined by the experimental z-stack data [30]. Please refer to Section 2.8 for more details on this modelling approach.

We considered the increase in web protein concentration inside the membranous web volume zone to be “activation” of the membranous web zone. In our framework, nonstructural proteins induced ER remodeling by detaching web proteins from the ER and facilitating their diffusion into the respective potential membranous web zones. We restricted web protein diffusion to the experimentally defined potential membranous web zones.

### 4.3. vRNA Movement and Location

Experimentally, the spatio-temporal dynamics of HCV RNA have not yet been fully described [50]. Mathematical modeling approaches enable the mimicking of different hypotheses and the prediction of typical scenarios following them.

In this context, we focus on two major properties: viral RNA location and transport.

Regarding location, it would be interesting to experimentally clarify whether viral RNA attaches to the ER surface, similar to nonstructural proteins, or if it moves solely in the cytosol volume or if both modes are possible. Our model incorporates both modes but can easily be restricted to one specific mode if experimental knowledge is required. Regarding transport mechanisms, both passive and active transport are theoretically possible. Typically, modeling begins with passive transport diffusion models, and we also follow this approach. However, for future work, active transport can be incorporated if there is experimental evidence supporting it.

Evaluating the presented simulations demonstrates that if viral RNA transport is based on diffusion, transport along the ER surface is much more efficient than transport in the cytosol volume. Diffusion in the volume leads to blurring of the viral RNA concentration, resulting in only a few viral RNA molecules reaching other ribosomal regions. Therefore, this form of transport appears to be inefficient in contrast to passive transport along the ER.

To enable efficient volume transport, it is tempting to expect the incorporation of active transport terms into the model, i.e., to incorporate advection terms in future studies and to analyze if such terms enable descriptions closer to future spatial experimental data. However, because such data are currently unavailable, we have not incorporated advection/active transport terms for viral RNA into the partial differential equation (PDE) model. Given that viral RNA volume diffusion is not efficient in our model, we propose that viral RNA polymerized in the membranous web volumes cannot directly attach to ribosomes. Instead, newly polymerized viral RNA diffuses in the membranous webs and in the cytosol but can attach to the ER surface. Once attached to the ER surface, free viral RNA diffusion becomes comparatively efficient. Free viral RNA can then diffuse to other ribosomal regions on the ER surface and can be bound as ribosomal viral RNA. Note that in our model, the total number of possible viral RNAs in a ribosomal region is restricted to a natural number, which we can select. In our example, we restricted this number to one.

We assumed that the cut set of the reconstructed ER surface and reconstructed membranous web boundaries indicate ribosomal regions and that the membranous web regions grow on top of these ribosomal regions. The restriction that a maximum amount of viral RNA can be bound per ribosomal region indicates that different sized ribosomal surfaces need to be able to supply different concentration maxima (suprema) of viral RNA. Our model achieves this by incorporating specific prefactors in the reaction term, which allows the capture of ER surface-attached free viral RNA to a ribosomal region, as described in ([Disp-formula FD7a-viruses-16-00840])–(7e).

In addition, our model incorporates the notion that NS5A may influence the diffusion of viral RNA, implying that NS5A aids in viral RNA transport. This property is optional and can be deactivated by setting $k_{1}=0$ and $k_{3}=0$ in ([Disp-formula FD7a-viruses-16-00840])–([Disp-formula FD8e-viruses-16-00840]).

Because we have introduced the concept of NS5A-enhanced (“boosted”) viral RNA transport in our qualitative surface model [32], we refer to this paper for a more detailed discussion on this topic. (Nevertheless, there is no need to study this reference to understand the major properties of this aspect.)

### 4.4. Replication Complex and the Cis Condition

The replication complex was modeled as a combination of web protein and previously ribosome-bound viral RNA.

This representation is reflected in the reaction coefficients of the equations for $W_{C}^{S}$ and $R_{R}^{S}$ of the surface partial differential equation (sufPDE) system ([Disp-formula FD7a-viruses-16-00840])–(7e) for detaching components. Similarly, in the partial differential equation (PDE) system, this is reflected through the boundary conditions ([Disp-formula FD9a-viruses-16-00840])–(9d) for $C_{W}^{V}$.

We previously introduced this concept (combination of web protein with viral RNA to form replication complex) in our earlier study [32]. In that model, since all processes were restricted to the ER surface, we were able to incorporate the combination of replication complexes in the formula of the analog of $C_{W}^{V}$ as a reaction term. However, because of the coupling of volume and surface effects in this study, this concept is implemented through the boundary conditions ([Disp-formula FD9a-viruses-16-00840])–(9d) of the PDE system ([Disp-formula FD8a-viruses-16-00840])–(8e).

As described in our previous study [32], the interaction between viral RNA and web protein in forming the replication complex embodies the “cis-requirement” [51,52,53] in the mathematical model. The cis condition stipulates that viral RNA can only be replicated by nonstructural proteins originating precisely from that viral RNA. Moreover, the fulfilling of the “cis requirement” by our simulation framework is not an input of our approach but rather a result.

Similarly, the spatial model proposed in this study describes that the web protein, as a cleaved part of the polyprotein, is translated by viral RNA bound at a specific ribosomal region. This web protein is unable to move to any membranous web zone except for the one belonging to this particular ribosomal region. Because the replication complex consists of web protein produced at this ribosomal region and viral RNA that had translated this web protein, each replication complex can only reproduce the viral RNA from which it was originated.

For more details on the “cis-requirement”, refer to Appendix B.

We emphasize that our model framework is, to the best of our knowledge, the only existing mathematical model of virus replication that mimics the cis-requirement.

### 4.5. Nonlinear Diffusion and Reaction Coefficient Structure

Nonlinear form of diffusion and reaction coefficients of the general form
(13)$$\frac{Q}{Q+p}$$
of an unknown concentration *Q* with a constant coefficient *p* enables the replacement of linear term structures that are unbound and realistic only for small concentrations [32]. This structure, as proposed in [32], allows for relatively sharp transitions between the deactivation of a function at low concentrations and activation at relevant concentrations. Moreover, it facilitates the establishment of a plateau value similar to the maximum value, for large concentrations.

This “maximum value” plays the role of the “maximum reaction velocity” that one would obtain for the Michaelis–Menten kinetics. (From a strict mathematical point of view, it is a supremum rather than a “maximum”).

The implementation of such terms instead of a constant value for the diffusion coefficients of the concentrations $W_{R}^{S}.W_{W}^{V}$ and *H* facilitates the modeling of the movement of web proteins and host factors within the membranous web zones when the web protein concentration is sufficiently high.

Moreover, the diffusion coefficients of the concentrations $N_{W}^{V},C_{W}^{V}$ enable the modeling of NS5A and replication complex entry into membranous webs only when the web protein concentration is present. Without web proteins in the membranous web subdomains, neither NS5A nor the replication complex can enter, reflecting their inability to enter a non-existing or inactive membranous web zone.

For further elucidations of nonlinear structured diffusion and reaction coefficients, we refer to [32].

### 4.6. Comparison to Quantitative Experimental Data

Based on experimental data, Dahari et al. [19] reported that the total amount of viral RNA and NS5B in a typical hepatoma cell after approximately 2 days of infection ranges from approximately $10^{3}$–$10^{4}$ particles for plus stranded viral RNA and approximately $10^{6}$ particles for NS5B. Given that all nonstructural protein types arise at the same quantity due to their cleavage from the polyprotein, neglecting decay processes, this magnitude can be adopted to each nonstructural protein type, namely web protein and NS5A, and, if we would model it, NS5B.

Comparing these data with our simulation results, which were conducted for approximately 1% to 10% of a cell and over a time scale of approximately half a day, our simulation data align with these experimental studies, cf. as detailed in Section 3.3.

### 4.7. Quantitative Model Validation—Spatial Patterns and Parameter Set

Validation of the spatial patterns in our simulation remains unfeasible to date. Spatially resolved experimental data will be essential for future adaption and validation of our model.

Overall, our model strikes a reasonable balance between incorporating crucial details and over-parametrization by including only those details that can be validated. This may be true even though a considerable number of parameters and complexity are still included.

Living matter often involves many parameters. However, this also holds true for every problem in which one takes into account interactions for which no measured data are available at the time in which the problem is formulated. What makes biological systems particularly difficult is, perhaps, the fact that they “force” the modeler or theoretician to conceive them in a way that includes several scales.

### 4.8. Relationship between Form and Function

Spatial models offer valuable insights into understanding the relationship between the form and function of the interaction between the virus replication machinery and its host–cell interface.

In our initial study in this context [31], we conducted spatial investigations. Even a simplified spatial model allowed us to reconsider and reinterpret the relationship between the host diffusion coefficient, host initial value, and overall viral RNA synthesis rate.

From the parameter sensitivity studies in this study, we concluded that viral RNA diffusion in the volume is significantly less effective than viral RNA diffusion at the ER surface (data not shown), regardless of the viral RNA diffusion coefficient in the volume. Therefore, this study supports the notion that passive viral RNA transport in the volume is unlikely to be the major source of viral RNA movement in the cell.

However, this statement remains valid only if viral RNA transport properties are governed by diffusion and are purely passive. If advective/active transport terms were to contribute, these results would change substantially. Investigation of these phenomena will be of interest for future studies, particularly in combination with experimental investigations.

### 4.9. Qualitative Model Properties

The qualitative previous properties of the model structure, in principle, do not differ from those described in the former study [32], where we asserted that the incorporation of volume effects would not alter the qualitative properties, but only the quantitative ones, and this assertion remains valid. Therefore, for an extended overview of the model structure beyond the explanations presented in this study and for possible model extensions concerning the general model structure, we refer to [32].

### 4.10. Aims of Our Work and Milestones

Considering experimental spatial data as “in vitro microscope screenshots” and simulation spatial data as “computer simulation screenshots”/“in silico microscope screenshots” (each for a fixed time point), we summarize the major aims of our work as follows:Establish mathematical equations describing the biophysics of intracellular virus replication in a fully 3D spatio-temporal resolved manner.Develop a computational framework capable of efficiently and robustly simulating these equations, which comprises:(a)Interpolation of experimental spatio-temporal virus replication states(b)Extrapolation of experimental spatio-temporal virus replication states to times beyond those already measured.(c)Prediction of spatio-temporal patterns of interactions among major components of intracellular virus replication.The 3D in silico “computer simulation screenshots”/states complement observable 3D “in vitro microscope screenshots”/states.Ensure the framework’s flexibility for extensions and adaptions. Particularly, for future applications it should enable in silico probing of the action of direct antiviral agents (DAAs) on spatio-temporal virus replication dynamics.

Our findings confirm the fulfilling of point 1 and the establishment of foundations for point 2 and 3. However, experimental spatial data are required to verify whether milestones 2 (a–c) and 3 have been met.

Despite this, significant progress has been made, with our simulation data aligning with available experimental data and exhibiting plausible spatial patterns. This advancement allowed us to solicit experimental data for validation, i.e., to characterize which further experimental data are needed for validation.

The current stage indicates significant progress, positioning the major components of our framework at an advanced stage. To answer specific biological questions, e.g., in the context of optimizing antiviral therapy, additional adjustments might be needed to reach milestone 3 completely and go forward to Point 4 in future work.

### 4.11. Summary: In Silico Microscope

Visualization of the computer simulations, performed on the HLRS Stuttgart Apollo Hawk supercomputer, resembles the look into some sort of “in silico microscope”.

In this paper, we demonstrated how our model recapitulates major experimental issues of the viral RNA replication cycle not only in qualitative but also in a quantitative manner.

The fully spatio-temporal simulations align with experimental observations, providing a glimpse into the dynamics of the viral RNA cycle similar to peering through a microscope into a real in vitro cell.

Moreover, the integrated values of the components across the computational domain serve as indicators for comparison with the existing data and kinetic compartment/ordinary differential equation (ODE) model values. The quantitative values of the temporal development of viral RNA and nonstructural proteins in the present study biologically align with values found in the literature based on experimental data and ODE model kinetic data [19].

Although our virus model simulation framework has reached a significant milestone, it still requires validation through important validation tests. While fulfilling the basic requirements outlined in Section 4.10, further improvements are necessary to address mid- and long-term challenges.

The in silico methods developed for HCV can be adapted for other RNA viruses, such as the closely related Coronaviridia family, including SARS-CoV-2 and MERS virus, and, in the middle term, also extended to other virus types such as retroviruses/DNA viruses.

A comprehensive understanding of virus replication mechanisms at the cellular level can reveal the fundamental processes that viruses use for self-replication. This knowledge could help in the development of effective direct antiviral agents (DAAs) capable of halting virus replication at its core—the single human and animal cells where virus replication occurs.

Furthermore, a detailed understanding of virus replication mechanisms may facilitate the development of efficient and long lasting vaccines, as they exist already for example for the Yellow fever virus [54]. The characteristics of the vaccine for the Yellow fever virus (YFV) [54], though serendipitous and not the product of systematic research, underscores the potential for informed vaccine development: One injection of YFV is sufficient to deliver life-long lasting complete immunity, and vaccinated-ones cannot transmit the disease any more [55]. Except for immune-suppressed persons, YFV vaccine is nearly free of side effects.

## 5. Conclusions

We conducted simulations of the HCV RNA replication cycle with full spatio-temporal resolution, incorporating effects occurring on the ER surface manifold and within the entire volume, with a special emphasis on the component exchange between the manifold and volume. The simulations were performed at realistic reconstructed geometries based on experimental fluorescence hepatoma cell data.

Although the overall number of viral RNA and nonstructural proteins in hepatoma cells align with experimental data, further validation is necessary, particularly through comparison with spatially resolved experimental data.

Despite these restrictions, our results present the first spatial model capable of providing plausible values for the number of viral components within individual hepatoma cells.

Our numerical framework enables visualization and evaluation of simulation data of mathematical partial differential equation (PDE) models describing intracellular virus replication. This evaluation technique can be considered as some sort of “in silico microscope”, complementing in vitro/in vivo research.

To facilitate the generation of quantitative model parameters for spatial virus models, we recently improved substantially the efficiency of estimating diffusion constants for intracellular membrane-bound components [34].

Our framework is adaptable beyond HCV, extending to other plus-stranded RNA viruses such as SARS-CoV-2, MERS, Yellow fever, and Dengue fever, and with some adjustments, to other virus types. This framework may significantly improve the systematic biophysical understanding of intracellular virus replication mechanisms. In addition, our framework aids in systematically planning intracellular virus research experiments to identify potential targets for virus elimination.

In the long term, a combined approach integrating in vitro/in vivo experiments with in silico simulations could expedite the design of cost-effective yet potent direct antiviral agents and the development of vaccines with minimal side effects, capable of inducing sterilizing immunity and long-lasting protection.

## Figures and Tables

**Figure 1 viruses-16-00840-f001:**
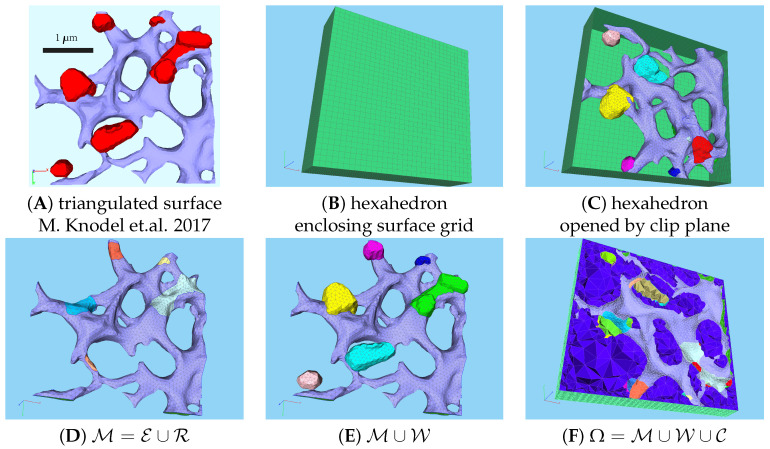
Computational domain generation for the partial differential equation (PDE) model to be developed in the forthcoming sections, and definition of its subdomains. (**A**–**C**) Volume mesh generation based on a given surface geometry: (**A**) Reconstructed surface grid describing the ER and membranous web (MW)surfaces. (Screenshot published first in our previous study [31]. ER surface in dark blue, MW surfaces in red) (**B**) Enclosure of the surface grid by a rectangular hexahedron (green) to allow tetrahedralization. (**C**) Hexahedron opened by a clip plane. (ER surface in dark blue, MW surfaces in different colors to account for their assignment to different subdomains, as they are not connected spatially; each MW subdomain is marked with its own color). (**D**–**F**) Volume mesh subdomains after tetrahedralization: (**D**) ER surface with ribosomes (ER surface in dark blue, ribosomes with different colors, as spatially not connected, to account for different subdomains), (**E**) MW (membranous web) volume regions (different subdomains refer to the different spatially unconnected MW zones) and ER surface (dark blue), (**F**) volume mesh opened by a clip plane (colors as in (**E**), but with additional color for the now visible cytosol subdomain). (Perspectives of the screenshots differ from each other in most cases).

**Figure 2 viruses-16-00840-f002:**
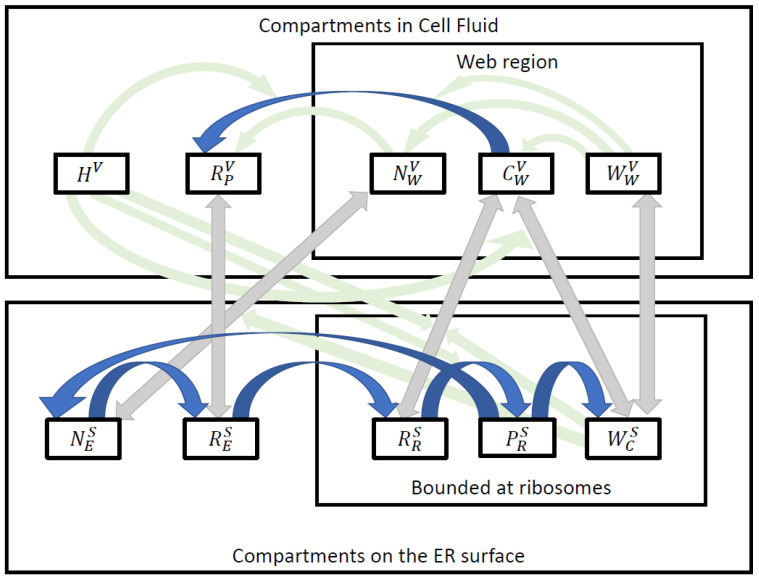
This simplified graphical representation of the partial differential equation model illustrates the interaction of the main components. It shows the model compartments in the white rectangles (see also Table 2) in the respective regions. The main processes are shown by blue arrows and transitions between the surface and volume by gray arrows: viral RNA $R_{R}^{S}$ bounded at ribosome induces translation of viral polyprotein $P_{R}^{S}$, which is cleaved in non-structural proteins $W_{C}^{S}$ and $N_{E}^{S}$. Non-structural proteins can detach from the ER surface and diffuse in the membranous web volume zone as $W_{W}^{V}$ and $N_{W}^{V}$. Replication complexes $C_{W}^{V}$ are formed, which produce polymerized viral RNA $R_{P}^{V}$. Viral RNA can attach to the ER and form $R_{E}^{S}$, which can again be bound to the ribosome to form new viral RNA $R_{R}^{S}$. Further compartments are involved in these processes and these involvements are indicated by light green arrows. This is especially true for the generic host factor $H^{V}$, which is involved in several processes.

**Figure 3 viruses-16-00840-f003:**
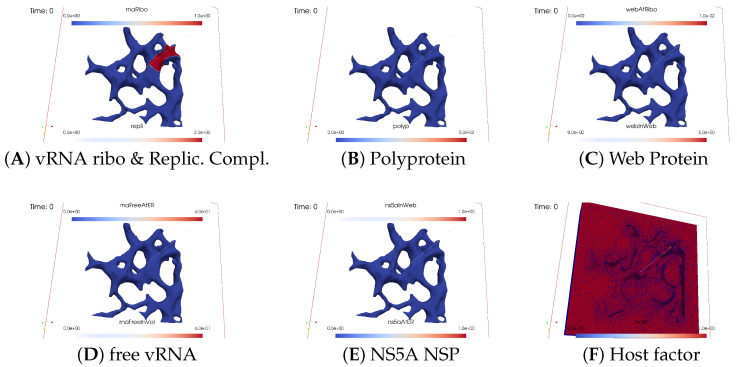
Initial state of the simulation, t=0: One vRNA (viral RNA) attached to one ribosomal region (**Panel A**); otherwise, the cell is healthy. All other viral components do not exist so far. In particular, so far, no “active” membranous web exists. The host factor is distributed homogeneously over throughout the cell. As in the following figures, high concentrations are indicated by dark-red color, low concentrations by dark-blue color.

**Figure 4 viruses-16-00840-f004:**
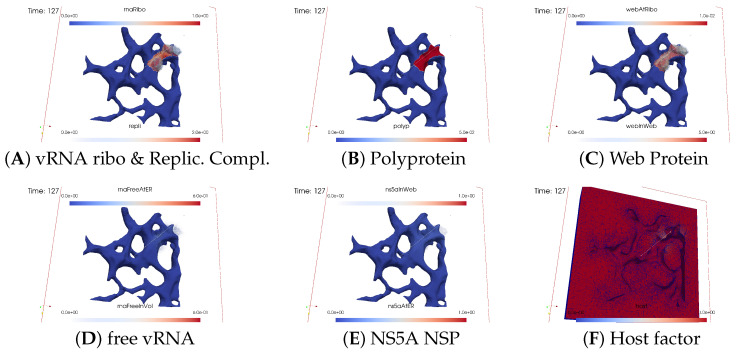
Simulation status at time t=127 s ≈2 min. The contamination of the cell by viral components starts, especially polyprotein translation followed by polyprotein cleavage. The web protein (WP) detaches from the ER surface and causes the growth of the first biophysically active membranous web zone, which is only weakly visible to date. The replication complex (RC) starts to emerge inside the membranous web as a combination of a previously ribosomally bound web protein and viral RNA (vRNA).

**Figure 5 viruses-16-00840-f005:**
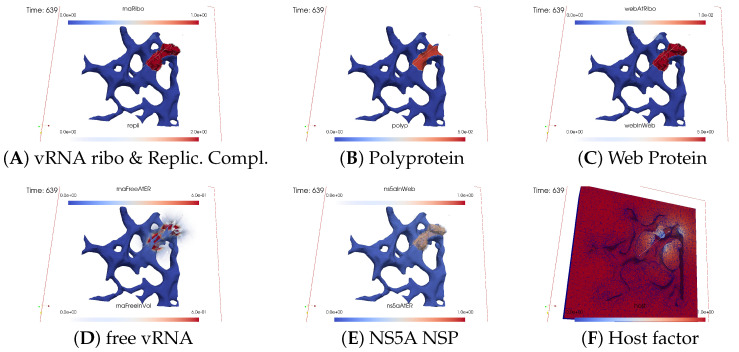
Simulation status at simulated time t=639 s ≈10 min. Begin of vRNA (viral RNA) polymerization in the first MW (membranous web), viral RNA volume diffusion, and viral RNA attachment to ER surface, followed by surface diffusion.

**Figure 6 viruses-16-00840-f006:**
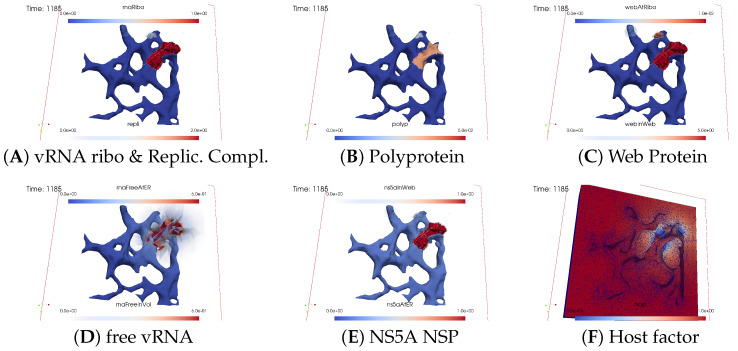
Simulation status at time t=1185 s ≈20 min. We can observe the closing of the viral RNA cycle: Newly polymerized viral RNA attached to the ER diffuses to the second ribosomal zone. Surface-bound viral RNA is captured by the next ribosomal zone once it arrives. The newly attached ribosomal viral RNA translates polyproteins. The polyproteins cleave into web protein and NS5A. The host factor is slightly reduced in the first membranous web zone.

**Figure 7 viruses-16-00840-f007:**
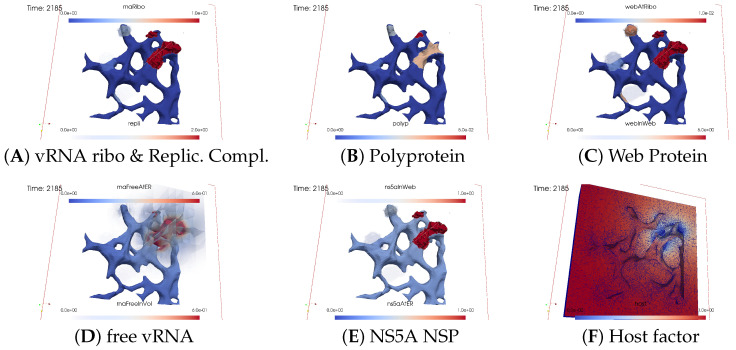
Simulation status at the simulated time t=2185 s ≈40 min. In the second ribosomal zone, the polyprotein is cleaved into the web protein (WP) and NS5A. The web proteins detach from the ribosomes, diffuse into the geometric membranous web (MW) subdomain volume, and induce further membranous web zone activation.

**Figure 8 viruses-16-00840-f008:**
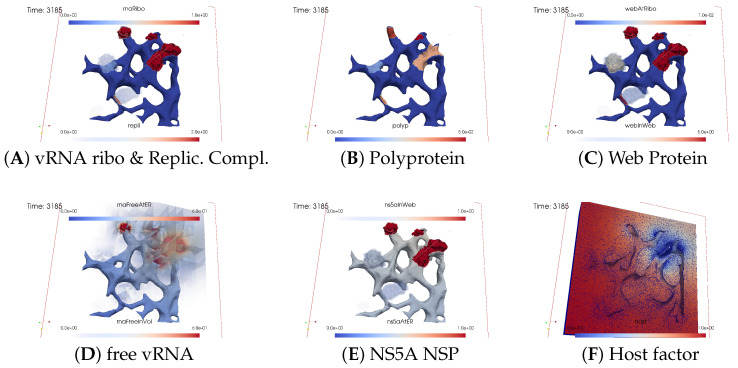
Simulation status at the simulated time t=3185 s ≈ 50 min. The viral RNA begins to rush through the cell. More membranous web zones become active and the host factor becomes depleted, initially slowly, then increasingly faster. The host factor also starts to reduce in the second and even third membranous web zone because of the delivery of energy for viral RNA polymerization.

**Figure 9 viruses-16-00840-f009:**
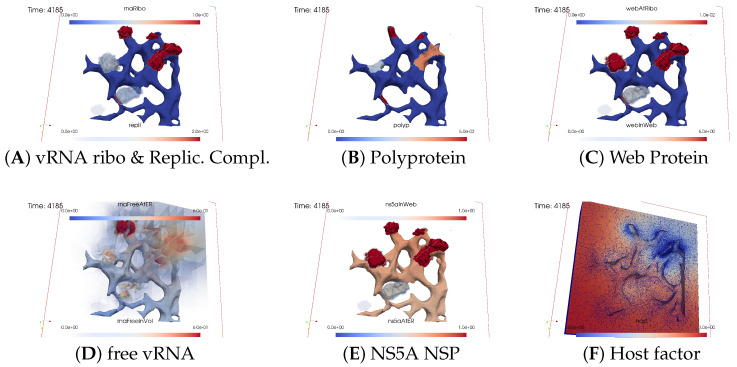
Simulation status at the simulated time t=4185 s ≈ 1 h 10 min. As viral RNA propagation, membranous web (MW) activation, and viral RNA polymerization continue, holes increasingly arise in the host factor, which still mirror the membranous webs but have already started to unify. While new spots arise where viral RNA is polymerized, former spots blur increasingly once the host factor is depleted and the surrounding cytosol is depleted as well. Replication complexes are entering more replication centers (active membranous web zones).

**Figure 10 viruses-16-00840-f010:**
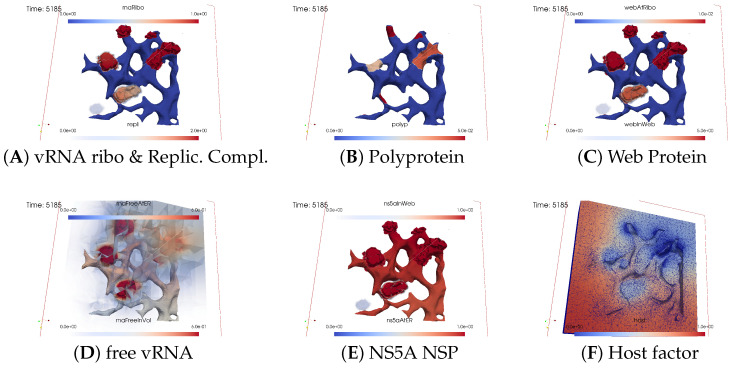
Simulation status at the simulated time t=5185 s ≈ 1 h 30 min. Membranous web activation continues. Most potential membranous web zones are activated and polymerize viral RNA. The membranous webs are the replication centers where the replication complexes polymerize viral RNA. Formerly active membranous webs increasingly encounter interruptions in viral RNA polymerization because of the lack of host factor.

**Figure 11 viruses-16-00840-f011:**
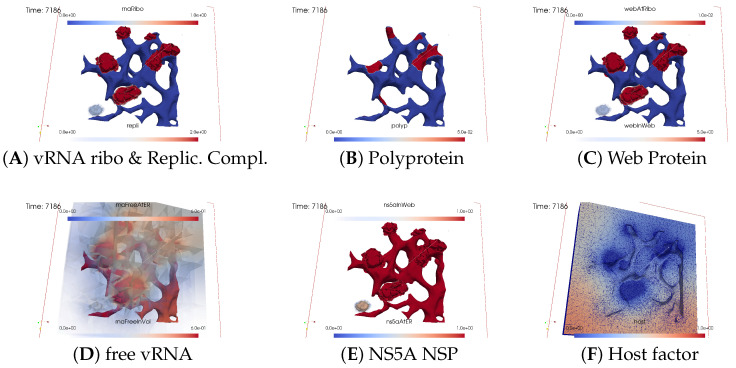
Simulation status at the simulated time t=7186 s ≈ 2 h. When viral RNA polymerization has stopped because of a lack of host factor, the membranous webs remain activated, but the viral RNA increasingly vanishes from the former active centers of replication because of diffusion and attachment to the ER surface. The host factor reduces globally, not only, but most strongly, in the replication centers. The last membranous web zone was activated and began to polymerize viral RNA.

**Figure 12 viruses-16-00840-f012:**
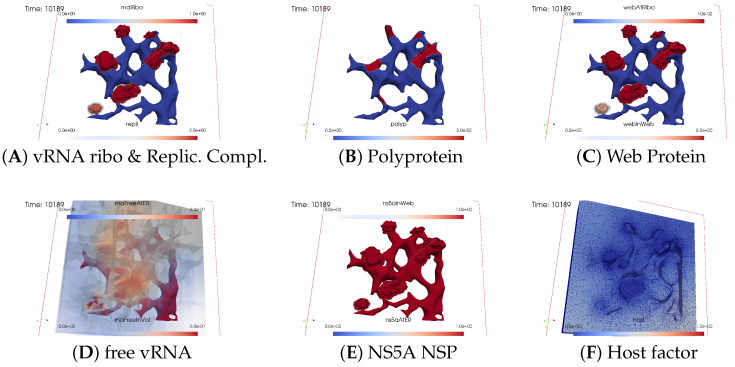
Simulation status at the simulated time t = 10,189 s ≈ 3 h. In addition, the last membranous web is now a viral RNA spot, while the host factor is so low that it starts to blur everywhere. All membranous web zones are now very strongly visible, but most of them have completed polymerizing viral RNA because of the lack of host factor.

**Figure 13 viruses-16-00840-f013:**
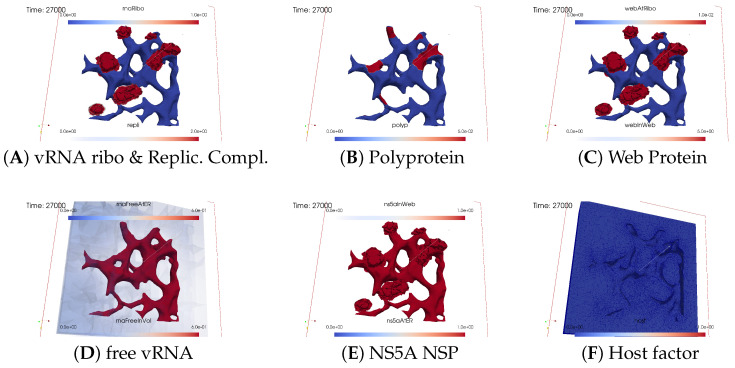
Simulation status at the simulated time t = 37,000 s = 7.5 h. The vast majority of the host factor is depleted, and most of the viral RNA is now located on the ER surface and within the replication complexes inside the membranous webs. Note that viral RNA is still everywhere in the cytosol volume and the membranous web zones, but is completely blurred compared with the initial state shown in Figure 3.

**Figure 14 viruses-16-00840-f014:**
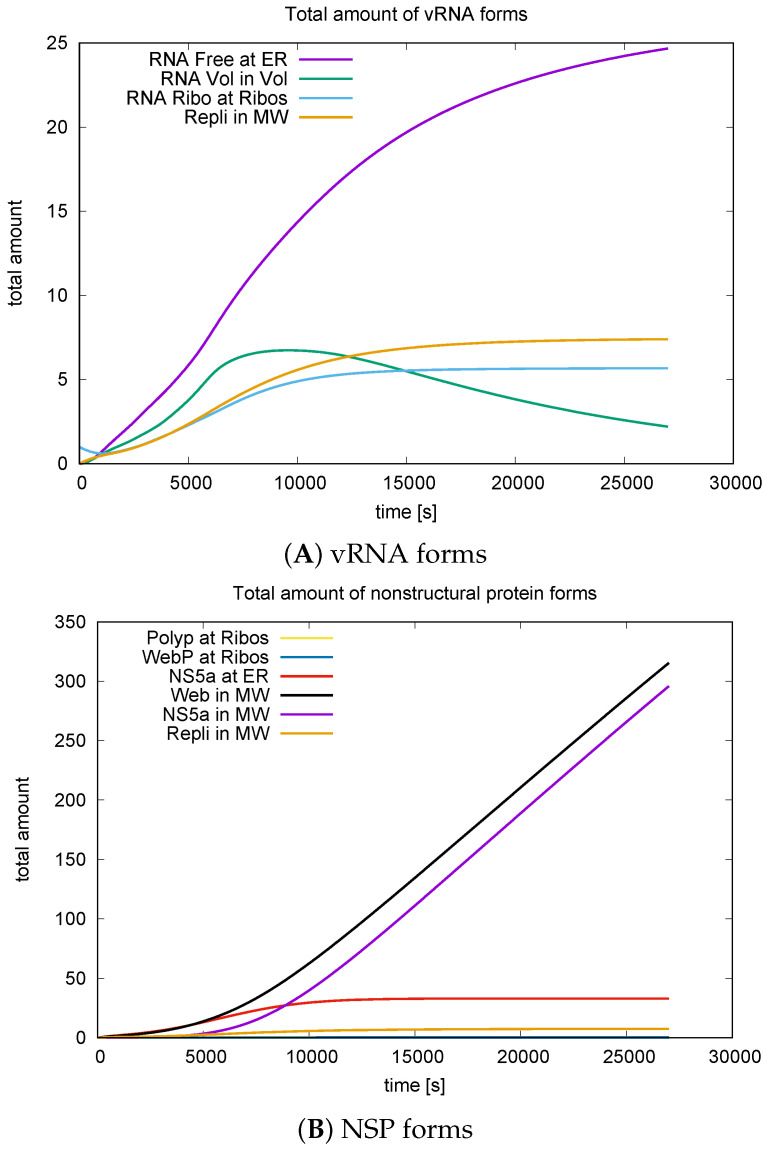
Long-term behavior of integrated compartmental concentrations in the complete computational domain (approximately 1% to 10% of a hepatoma cell) over time. Total simulated biophysical time approximately 7.5 hours. Results are computed at grid refinement level 4. (Note that those lines which are noted in the legends but not visible in the plots have such small values, that it effectively impossible to distinguish them from the x axis by blank eye vision.)

**Table 1 viruses-16-00840-t001:** Subdomains of the computational domain Ω. The merged surface-volume grid is built from the ER surface, ribosomal surfaces, and membranous web and cytosol volumes. It is enclosed by the hexahedron B.

Subdomain	Property
2D manifold E∪R=M⊂Γ, embedded in 3D	
E⊂M	reconstructed ER surface except for
$\cup_{i=1}^{7}R_{i}=R\subset M$	7 ribosomic zones: intersection ER/MW surfaces
2D closure of computational domain - hexahedron B	
B=∂Ω∖M	surface of enclosing hexahedron
3D volume C∪W⊂Ω	
C⊂Ω	cytosol (enclosed by the box B enclosing in fact Ω)
$\cup_{i=1}^{7}W_{i}=W\subset \Omega$	7 MW subdomains: volume enclosed by MW surfaces

**Table 2 viruses-16-00840-t002:** Compartments (concentrations) considered in the PDE (partial differential equation) model. Note that the surface concentrations are given in $\frac{1}{\mu m^{2}}$, whereas the volume concentrations are given in $\frac{1}{\mu m^{3}}$.

Compartment	Region	Biophysical Interpretation
Concentration		
Surface compartments (concentrations) defined at the 3D embedded curved 2D manifold M		
$R_{R}^{S}$	R	Ribosomal bound vRNA
$P_{R}^{S}$	R	Viral polyprotein translated at ribosomes
$W_{C}^{S}$	R	Web (NSP) protein/WP cleaved from the polyprotein
$N_{E}^{S}$	M	NS5A NSP cleaved from the polyprotein
$R_{E}^{S}$	M	polymerized free vRNA attached to the ER
Volume compartments (concentrations) defined in the 3D volume Ω		
$W_{W}^{V}$	W	Web (NSP) protein/WP detached from ribosomes to form MWs
$N_{W}^{V}$	W	NS5A NSP detached from ribosomes incorporated into MW
$C_{W}^{V}$	W	Replication complex/RC as combination of detached $R_{R}^{S}$ and $W_{C}^{S}$
$R_{P}^{V}$	Ω	Polymerized free vRNA moving in the full volume
$H^{V}$	Ω	Host factor

**Table 3 viruses-16-00840-t003:** Numerical values for the parameters of the model used in this study. It is noteworthy that while we have drawn inspiration from the literature for the magnitude of these parameters, we do not strictly adhere to the exact literature values. These values are primarily derived from ODE models and cannot be directly applied to our PDE-based model. Specifically, the reference for $r_{2}$ is [19], and for $D_{N}$ is based on (the PDE computations reasoned) [33].

Parameter	Value	Unit
δ	$1\cdot 10^{-2}$	μm^2^
$r_{1}$	0.001	$\frac{1}{s}$
$r_{2}$	100./3600./10. = 0.0028	$\frac{1}{s}$
$r_{3}$	0.056	$\frac{1}{s}$
$r_{4}$	0.1	$\frac{1}{s}$
$r_{5}$	0.05	$\frac{1}{s}$
$r_{6}$	0.002	$\frac{1}{s}$
$r_{7}$	0.005	$\frac{1}{s}$
$r_{8}$	0.001	$\frac{1}{s}$
$D_{R}^{S}$	0.001	$\frac{\mu m^{2}}{s}$
$D_{P}$	0.005	$\frac{\mu m^{2}}{s}$
$D_{N}$	0.01	$\frac{\mu m^{2}}{s}$
$D_{C}$	0.0005	$\frac{\mu m^{2}}{s}$
$D_{R}^{V}$	0.0001	$\frac{\mu m^{2}}{s}$
$D_{H}$	0.0005	$\frac{\mu m^{2}}{s}$
$p_{1}$	0.01	1
$p_{2}$	0.01	1
$p_{3}$	$1\cdot 10^{-8}$	1
$p_{4}$	0.01	1
$p_{5}$	0.01	1
$p_{6}$	0.01	1
$p_{7}$	0.01	1
$p_{8}$	0.01	1
$p_{9}$	0.01	1
$p_{10}$	0.01	1
$k_{1}$	1	1
$k_{2}$	1 $\vec{x}\in W$, 0 else	1
$k_{3}$	1 $\vec{x}\in W$, 0 else	1
ν	1	1
$v_{1}$	1	1
$v_{2}$	1	1
b	1/3	1
$h_{0}$	1	$\frac{1}{\mu m^{3}}$

**Table 4 viruses-16-00840-t004:** Degrees of freedom (DoF) and the number of tetrahedral volumes at different grid levels.

Level	DoFs	Vols
0	96,370	41,446
1	667,280	331,568
2	4,890,410	2,652,544
3	37,266,790	21,220,352
4	290,579,070	169,762,816

## Data Availability

Data will be shared with interested scientists on demand.

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
