# Peer review of "Intracellular “In Silico Microscopes”—Comprehensive 3D Spatio-Temporal Virus Replication Model Simulations"

_viruses, 2024, doi:10.3390/v16060840_

Round 1
Reviewer 1 Report
Comments and Suggestions for Authors
Please see the attached PDF file.

Comments on the Quality of English LanguageSome sentences should be improved. Some typos should be corrected.
Author Response
Please find the point to point analysis attached as pdf.

Reviewer 2 Report
Comments and Suggestions for Authors
The proposed work models the 2D and 3D distribution of reagents in a cell, with a realistic geometry of 2D surface, with the potential comparison with high resolution experiments. The topic is important.
Major comments
The factors that make the work impenetrable:
(i) poor English (see a sample below)
(ii) the assumption that the reader knows the previous work of the authors
(iii) the lack of definition of the biological questions addressed
(iv) the lack of schematic model diagrams showing the viral replication cycle and the biological processes under study
(v) one-line figure captions, that should be much longer
(vi) model equations come too late when the reader already gave up,
(vii) lack of justification of the complexity of the model given in the end (Discussion or before), with reference from Model section. Given the question they address, can they simplify the model or not?
I think these features must be fixed before sending the manuscript to an expert on 3D modeling. I cannot judge the validity results.
Please enlist the help of a colleague more proficient in literary English. The entire text needs rewriting.
Minor comments:
Line 4: “is inevitable “. If it is inevitable, we do not need to do research. Why bother writing a paper? May be, “necessary”?
Line 4: “form and function” of what?
Line 5: “(DAA)”: Please do not use any abbreviations in Abstract.
Line 7, 8: “fully spatio-temporal 7 resolved virus replication dynamics simulations” replace with “simulations of this dynamics”.
Line 10: Remove “advanced highly” and “genome” as not bearing information and “the” in the next line.
Line 12: remove “RNA (vRNA)” and explain “vRNA” in the end: is it genomic RNA or mRNA or both?
Line 13: The readers of Viruses never heard of PDE or even ODE in their entire lives. Use full names, as for every single abbreviation.
Line 14: “Endoplasmic”
Line 15: The sentence does not make any sense. Are these replication factories growing after the rain, like mushrooms?
Line 16, 27: at-> for, “which are” are redundant. “reconstructed” from what? From data? Say so.
Line 19: “put on”.
Line 22: The sentence carries no information and is written in informal language.
Line 41: This is not true. Only roughly 40% of chronic liver disease is caused by HCV, and another roughly 40% are by alcohol and substance abuse. The remaining ~ 20% are due to various congenital diseases.
Line 45: replace DAA with “below referred to as “drugs””. Less abbreviations is better.
Lines 47-53: these sentences do not make a slightest sense. I cannot even guess what you meant.
Line 55-57 should read: “Dynamic mathematical models [16–24] using ordinary differential equation (ODE) have been developed for HCV infection to determine the optimal doses of DAAs and other antiviral agents.”.
Line 58: remove “cf. e.g. “.
Lines 61-62: what quantities are “spatially-resolved”?
Replace “highly spa tially resolved” with “spatially-resolved”, because “highly” is not defined.
Line 66: what is “remodeling processes” in ER?
Line 67: remove “More in detail but still very superficial”
Line 20: “vRNA spatial properties” are unclear. Do you mean spatial distribution? Of what viral product, genomic RNA?
Line 70: “speaking”. It is more clear for biologists to write “represents a system of connected tori, with their total number g, where...”.
Line 73-75: remove “the very advanced experimental data such as highly “ and “RNA (vRNA)”. They do nor carry information and confuse.
Line 75: “entire nature of kinetic”--> “limitation”.
Line 76-77: replace “As kinetic ODE models by nature” with “they” and remove “there is space for 76 additional techniques”. They carry no information and do not help the flow.
Section 1.2 is long, confusing, and totally unnecessary. Save it for your Nobel prize speech.
Line 80-81 is unclear. What is in silico microscope like? What parameter is spatiallu resolved?
Lines 87-90: The two sentences do not make sense, in the context.
Lines 90-92 and 93-94 are in no logical connection either.
Lines 97-98 are impossible to understand. Please define the terms and fix syntax.
Line 106: Form and function of what?
Line 107: velocity is NOT diffusion coefficient
Line 108: what is “RNA genesis?”
Line 110: What are “modes of action”?
Line 111: You mean “mass action law”? The rates or processes proportional to amounts of reagents? Say so.
Line 112-113 does not make any sense.
(I skip the rest of 1.2, it has to be removed)
line 134-135. Specify all acting reagents and their location (2d and 3d)
line 135-140: This has to go to Model section. Say in one sentence “we take kinetic coefficients from the previous modeling or experimental work”—and cite the specific papers.
lines 141-143: unclear
lines 144-147: Say plainly “Our work is designed to interpret experiments”—and explain WHAT quantity they measured, not only how.
lines 184-189: unclear
Fig. 1 has to be cited the first time you mentioned data. The legend has to be much longer and describe what is shown.
Section 2.1 is impenetrable, because the authors assume that the reader has studied their previous work.
Comments on the Quality of English LanguageSee above.
Author Response

(The authors gave the same response as above.)

Round 2
Reviewer 1 Report
Comments and Suggestions for Authors
I have gone through the revised manuscript and suggest its publication in its present, revised form.
Reviewer 2 Report
Comments and Suggestions for Authors
The authors have made a strong effort to address my comments.
Comments on the Quality of English LanguageMajor improvement. I only found a few typos like "undergoe" instead of "undergo", or odd expressions like "initiate the existence" instead of "emerge".